# Mechanisms and targets of Fcγ-receptor mediated immunity to malaria sporozoites

Gaoqian Feng [1,2], Bruce D. Wines [1,3,4], Liriye Kurtovic[1,3], Jo-Anne Chan [1,3], Philippe Boeuf[1,2], Vanessa Mollard [5], Anton Cozijnsen[5], Damien R. Drew[1], Rob J. Center[1,2], Daniel L. Marshall[1,3], Sandra Chishimba [1,2], Geoffrey I. McFadden[5], Arlene E. Dent[6], Kiprotich Chelimo[7], Michelle J. Boyle [1,8], James W. Kazura[6], P. Mark Hogarth[1,3,4] & James G. Beeson [1,2,3,9 ✉]

A highly protective vaccine will greatly facilitate achieving and sustaining malaria elimination. Understanding mechanisms of antibody-mediated immunity is crucial for developing vaccines with high efficacy. Here, we identify key roles in humoral immunity for Fcγ-receptor (FcγR) interactions and opsonic phagocytosis of sporozoites. We identify a major role for neutrophils in mediating phagocytic clearance of sporozoites in peripheral blood, whereas monocytes contribute a minor role. Antibodies also promote natural killer cell activity. Mechanistically, antibody interactions with FcγRIII appear essential, with FcγRIIa also required for maximum activity. All regions of the circumsporozoite protein are targets of functional antibodies against sporozoites, and N-terminal antibodies have more activity in some assays. Functional antibodies are slowly acquired following natural exposure to malaria, being present among some exposed adults, but uncommon among children. Our findings reveal targets and mechanisms of immunity that could be exploited in vaccine design to maximize efficacy.

[1] Burnet Institute for Medical Research and Public Health, Melbourne, VIC, Australia. [2] Department of Medicine and Doherty Institute, The University of Melbourne, Melbourne, VIC, Australia. [3] Central Clinical School, Monash University, Melbourne, VIC, Australia. [4] Department of Clinical Pathology, The University of Melbourne, Melbourne, VIC, Australia. [5] School of BioSciences, The University of Melbourne, Melbourne, VIC, Australia. [6] Centre for Global Health and Diseases, Case Western Reserve University, Cleveland, OH, USA. [7] Department of Biomedical Science and Technology, Maseno University, Kisumu, Kenya. [8] Department of Immunology, IMR-Berghofer Institute, Herston, QLD, Australia. [9] Department of Microbiology, Monash University, Melbourne, VIC, Australia. ✉email: beeson@burnet.edu.au

Malaria remains a major global health concern, with *P. falciparum* malaria causing ~219 million clinical cases and 435,000 deaths annually[1]. Despite international control efforts, the malaria burden has remained constant in recent years[1]. There is a strong need for a highly efficacious vaccine for malaria control and elimination, which is further emphasised by increasing reports of antimalarial drug resistance and insecticide resistance[1]. The World Health Organisation and funding partners set a goal to develop a vaccine with ≥75% efficacy by 2030[2], but this has proven challenging to achieve[3,4]. A greater understanding of the mechanisms that mediate immunity is needed to develop strategies that generate more potent protective immune responses.

Malaria infection starts from an infected *Anopheles* mosquito bite, during which sporozoites are inoculated into the dermis[5]. Sporozoites then migrate to a blood vessel, circulate in the bloodstream to the liver and establish infection in hepatocytes. Sporozoites represent a priority target of malaria vaccines because clearing sporozoites will halt infection prior to the onset of clinical malaria[3]. Research in animal models and vaccine trials has established that antibodies are the key mediator of immunity to sporozoites (reviewed in ref. [3]). However, how antibodies function mechanistically to clear sporozoites and prevent infection, and the specific targets of functional antibodies, is not well understood. New insights on these questions are needed to facilitate the development of more efficacious vaccines.

The circumsporozoite protein (CSP) is the most abundant protein on the surface of sporozoites[6] and a major target of human antibodies[6]. CSP-based vaccines have outperformed other vaccine antigens to date, although efficacy remains sub-optimal[3]. The most advanced malaria vaccine, RTS,S, is based on a truncated form of CSP (containing only the central repeat region and C-terminal region) and achieved modest efficacy against clinical malaria (29–36%) in phase III clinical trial[7]. The central repeat region of CSP is considered an important antibody target, and consequently, the potential importance of antibodies to non-repeat regions has been vastly understudied[8], but may be important. Antibodies to the C-terminal region were associated with protection in RTS,S vaccine trials in children[9]. In terms of antibody functional mechanisms, the majority of published studies have focused on the direct inhibitory activity of antibodies to the central repeat region of CSP, which can inhibit sporozoite motility and hepatocyte invasion in vitro[10–17]. Little is known about other potentially important mechanisms and targets of immunity.

A key immunologic mechanism that is being increasingly recognised in immunity against viral and bacterial pathogens[18–21] is the ability of antibodies to interact with Fcγ receptors (FcγR) on various cell types leading to opsonic phagocytosis or direct cytotoxicity. Antibodies to sporozoites antigens are typically dominated by the IgG1 and IgG3 subclasses[13], which have strong potential to interact with FcγRs[22]. However, there is currently little known about these potential mechanisms in immunity to sporozoites. Major phagocytic cells types include monocytes and neutrophils (2–10% and 45–70% of peripheral white blood cells, respectively). In addition to phagocytosis, neutrophils and natural killer (NK) cells can also kill cells through antibody-dependent cellular cytotoxicity (ADCC)[23,24], and both monocytes and neutrophils also have wider immune impacts through the expression of activation markers, cytokines and chemokines (reviewed in refs. [25–27]).

FcγR expression patterns differ between monocytes and neutrophils, which impacts their functions in immunity. Since sporozoites represent the first step in initiating malaria infection, sporozoites would typically encounter potential phagocytic cells in a resting, or homoeostatic, state rather than an activated state. Resting-stage monocytes predominantly express FcγRI and FcγRIIa, with a small subset expressing FcγRIIIa[28], whereas resting-stage neutrophils express FcγRIIa and FcγRIIb[29], and low levels of FcγRIIIa[30] (which is important functionally). Evaluating specific interactions between antibodies and different FcγRs is therefore important for understanding the potential roles of different phagocytes in immunity. IgG subclass, glycosylation and epitope specificity can each impact on interactions with different FcγRs[22].

Defining FcγR-mediated mechanisms targeting sporozoites may reveal insights that could be exploited to enhance the protective efficacy of sporozoite-based vaccines. In this study, we reveal important mechanisms and targets of immunity against sporozoites mediated by specific FcγRs on immune cells. We develop techniques to measure interactions between antibodies and specific FcγRs, and identify specific antibody properties, antigenic targets and FcγRs that are important for optimal functional activity. Furthermore, we quantify the acquisition of functional FcγR-dependent antibody responses in children and adults naturally exposed to malaria. Our findings reveal important immune mechanisms against sporozoites and suggest strategies and antigenic targets for developing a highly efficacious malaria vaccine.

## Results

**Opsonic phagocytosis of sporozoites is predominantly mediated by neutrophils in peripheral blood.** To investigate the relevance of FcγR-mediated mechanisms against sporozoites, we first established opsonic phagocytosis assays using the undifferentiated THP-1 cell line (pro-monocytic cells), under standard conditions[31,32] (Supplementary Fig. 1A). To opsonize sporozoites, we used serum antibodies from Kenyan adults resident in a region with high malaria transmission intensity and have significant clinical immunity to malaria and antibodies to CSP[33]. Opsonized *P. falciparum* sporozoites were effectively phagocytosed by THP-1 cells in a concentration-dependent manner with little phagocytosis using antibodies from malaria-naive donors (Melbourne residents) (Supplementary Fig. 1B). Opsonic phagocytosis mediated by individual sera was highly reproducible ($r = 0.925$, $P < 0.001$; Supplementary Fig. 1C). To overcome the challenges in generating large numbers of viable sporozoites for opsonic phagocytosis assays, we established the use of fluorescent beads coated with full-length CSP, the major antigen expressed on sporozoites, as a surrogate measure of opsonic phagocytosis of sporozoites. CSP-coated beads opsonized with a selection of antibody samples from different Kenyan adults were effectively phagocytosed by THP-1 cells, and opsonic phagocytosis of CSP-coated beads strongly correlated with opsonic phagocytosis of *P. falciparum* sporozoites ($r = 0.831$, $P = 0.0015$, Supplementary Fig. 1D). Therefore, CSP beads provided an efficient platform for studying mechanisms and targets of antibody-mediated opsonic phagocytosis.

Subsequently, we investigated cell types that mediate opsonic phagocytosis in a whole-leukocyte assay with fresh blood, initially using CSP beads opsonized with polyclonal rabbit antibody to CSP. Interestingly, the rate of phagocytosis (number of beads phagocytosed per cell) was much higher by neutrophils than monocytes (Fig. 1A, B). Because we quantified the phagocytosis rate, the higher activity of neutrophils is not simply explained by their higher abundance compared to monocytes. Opsonic phagocytosis increased with increasing antibody concentrations, and higher relative activity of neutrophils compared to monocytes was seen across antibody concentrations and across a range of bead:cell ratios. The greater phagocytosis activity of neutrophils was confirmed using freshly isolated live sporozoites (we used

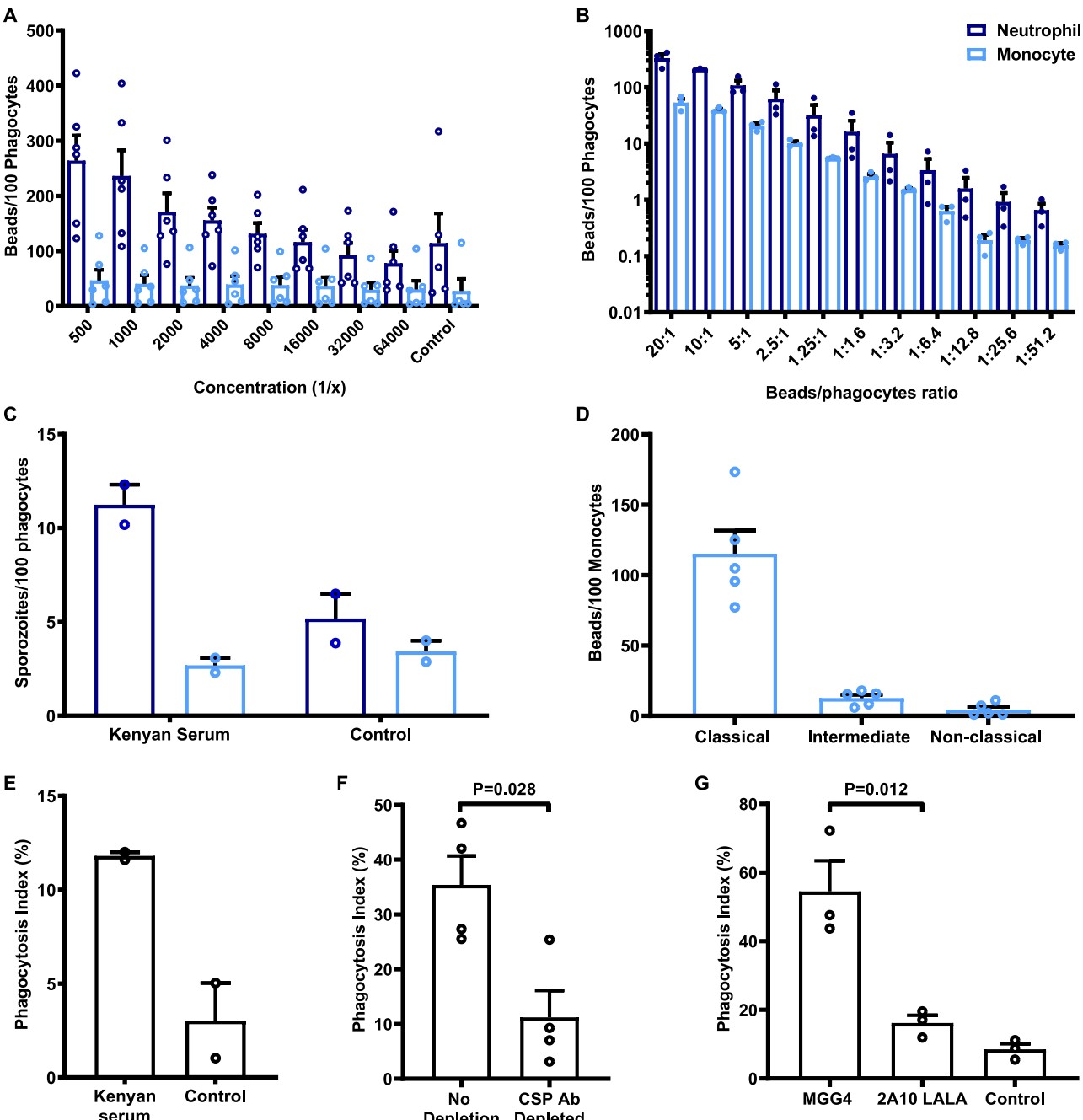

**Fig. 1 Neutrophils predominately mediate opsonic phagocytosis of CSP-coated beads and sporozoites in whole blood. A** CSP-coated beads were opsonized with different concentrations of rabbit anti-CSP IgG. Neutrophils showed greater phagocytosis activity compared to monocytes ($P < 0.001$, two-way ANOVA). Bars represent the mean and standard error from six independent experiments. **B** CSP-coated beads opsonized with rabbit anti-CSP IgG (1/500 dilution), were co-incubated at different ratios with whole leukocytes. Phagocytosis activity was substantially higher by neutrophils than monocytes ($P < 0.001$, two-way ANOVA). Bars represent the mean and standard error from three independent experiments. **C** Phagocytosis of sporozoites was tested using a transgenic PfCSP-*P. berghei* sporozoite line, opsonized with pooled immune sera from Kenyan adults. In the presence of immune antibodies, more sporozoites were phagocytosed by neutrophils (dark-blue bar) than monocytes (light-blue bar). Bars represent the mean and range from two independent experiments. **D** The number of beads phagocytosed by each monocyte subset was quantified and standardised as beads per 100 monocytes. The classical monocyte subset showed greater phagocytic activity ($P < 0.001$, one-way ANOVA). Bars represent the mean and standard error from five experiments. **E** Opsonization of *P. falciparum* sporozoites with a pool of malaria-exposed Kenyan adult sera induced a higher level of phagocytosis by neutrophils compared to unopsonized control. Opsonic phagocytosis activity was expressed as the percentage of ethidium bromide-positive cells. Bars represent the mean and range from two experiments. **F** Effect of depleting CSP-specific antibodies from a pool of malaria-exposed Kenyan adult sera on opsonic phagocytosis of *P. falciparum* sporozoites by neutrophils. Depletion of CSP-specific antibodies reduced opsonic phagocytosis by 70% (Man–Whitney test, $P = 0.028$). Bars represent the mean and standard error from four experiments. **G** mAb MGG4 (IgG1) was isolated from an individual exposed to *P. falciparum* sporozoites and was tested for promoting opsonic phagocytosis by neutrophils. Opsonic phagocytosis was significantly higher compared to a control mAb (2A10), which binds to the same epitope, but was modified (LALA mutations in the Fc region) to reduce FcγR binding (Mann–Whitney test $P = 0.012$). Bars represent the mean and standard error from three experiments.

transgenic *P. berghei* sporozoites that express *P. falciparum* CSP, which replaces the endogenous *P. berghei* CSP gene[12], referred to as PfCSP-*P. berghei*). Opsonic phagocytosis of sporozoites opsonized with antibodies from Kenyan malaria-exposed donors was predominately mediated by neutrophils with only low levels of phagocytosis by monocytes, and neutrophils showing a higher phagocytosis rate (Fig. 1C). We also performed assays to allow an extended time for phagocytosis (up to 60 min), but we still observed much greater phagocytosis by neutrophils compared to monocytes (Supplementary Fig. 2A). Further analysis of the monocyte population indicated that the majority of phagocytosis was by the classical subset (CD14high CD16−; Fig. 1D and Supplementary Fig. 2B–D). To support our findings, we further showed that purified neutrophils could effectively phagocytose antibody-opsonized *P. falciparum* sporozoites (Fig. 1E). Depletion of CSP-specific antibodies, by pre-incubation with recombinant CSP, reduced phagocytosis of sporozoites by 70% compared to the non-depleted sample (Fig. 1F and Supplementary Fig. 3). We used an additional approach to obtain evidence of the importance of CSP as a target of acquired functional antibodies in human immunity by testing a mAb (MGG4) isolated from an individual exposed to repeated sporozoite infections[34]. The mAb induced substantial phagocytosis of sporozoites, whereas there was little phagocytosis with non-immune human antibodies, or using an anti-CSP MAb (2A10) that was mutated in the Fc region to inhibit FcγR interactions (Fig. 1G).

**Defining Fcγ receptors involved in opsonic phagocytosis of sporozoites**. We opsonized whole *P. falciparum* sporozoites and CSP beads with antibodies from Kenyan adults and measured opsonic phagocytosis by neutrophils in the presence of specific FcγR-blocking antibodies. Neutrophils had significantly lower opsonic phagocytosis of both CSP beads and *P. falciparum* sporozoites in the presence of either FcγRIIa or FcγRIII blockers (Fig. 2A, B and Supplementary Fig. 4A), suggesting that both receptors were important. Notably, FcγRIII blockade was the most effective at inhibiting phagocytosis of *P. falciparum* sporozoites or CSP beads, with a weaker effect of FcγRIIa blockade. Opsonic phagocytosis by neutrophils was unaffected by blocking FcγRI. Titrating the concentration of blocking antibodies suggested we had achieved the maximal effect for the blockade of each individual FcγR (Supplementary Fig. 4B). Therefore, we combined FcγRIIa and FcγRIII blocking antibodies and found this gave greater inhibition of phagocytosis than using either blocking antibody, further suggesting the importance of both FcγR types for optimal phagocytosis (Fig. 2B). These results are consistent with the expression of FcγRs on resting neutrophils, which predominantly express FcγRIIa and FcγRIIIa/b, but lack FcγRI[35] (Supplementary Fig. 2C). The importance of FcγRIII in mediating opsonic phagocytosis of CSP beads and sporozoites may partly explain the relatively low activity of monocytes since the major monocyte subsets express little FcγRIIIa on the surface in a non-activated state and do not express FcγRIIIb[35]. THP-1 cells are widely used as a model of monocyte phagocytosis and highly express FcγRI[36], which is bound by monomeric IgG (whereas FcγRIIa and FcγRIII expressed by neutrophils only bind to immune complexes). Accordingly, we found that opsonic phagocytosis by THP-1 cells in standard conditions (containing 10% FCS only and no human serum) was strongly inhibited by the FcγRI blocker with limited inhibition by the FcγRIIa blocker, and none with FcγRIII blocking (Fig. 2C). This suggested that the interaction between IgG and FcγRI was particularly important for THP-1 cells, contrasting results seen with neutrophils. Therefore, we modified the standard THP-1 assay to make it more physiologically relevant by including human serum in the assay media.

In the presence of additional 2.5% human serum (as used in the whole leukocyte and neutrophil assays), there was much less opsonic phagocytosis (Fig. 2D), presumably due to the binding of non-immune monomeric human IgG present in the human serum to FcγRI. These findings may further explain the low phagocytic activity of primary monocytes observed in the whole leukocyte assays in the presence of human serum; bovine IgG (present in FCS) does not bind human FcγRI[37].

Given the apparent importance of FcγRIII, we further measured the function of antibodies to CSP among Kenyan adults in ADCC assays, since FcγRIII interactions play a key role in mediating ADCC via NK cells. We tested a selection of samples from the Kenyan adult cohort (Kanyawegi cohort) to represent high, intermediate and low responders (n = 31) in an assay using an ADCC reporter cell line that only expresses FcγRIIIa. Naturally acquired antibodies among Kenyan adults had significant activity, whereas antibodies from Melbourne malaria-naive donors did not (Fig. 2E, F). These findings suggest that antibodies to CSP may also promote ADCC activity through engaging FcγRIIIa. Subsequently, we tested the activity of primary NK cells in fresh blood using an established degranulation assay[38]. Incubation of NK cells with CSP beads opsonized by Kenyan adult antibodies leads to degranulation of primary NK cells (Fig. 2G, H). Activity varied across samples tested, and significantly correlated with FcγRIII binding (r = 0.538, P = 0.002, Fig. 2H), but weakly correlated with IgG to CSP (r = 0.127, P = 0.489, n = 31).

**Direct binding of FcγRIII and FcγRIIa by antibody–antigen complexes**. We developed FcγR-binding assays using FcγRIIa and FcγRIII that were expressed and purified as dimers so they can interact with antigen–antibody complexes[39]. Kenyan adults had a high prevalence and magnitude of antibodies to CSP that could interact with FcγRIIa and FcγRIII dimers (Fig. 3A). Overall, 67.3 and 61.5% of samples (n = 104) were considered positive for FcγRIIa and FcγRIII binding. While the binding of both FcγRs had a strong, positive correlation among samples (r = 0.71, P < 0.0001, n = 104) (Fig. 3B), some individual samples demonstrated high binding to one FcγR and low binding to the other. Furthermore, the ability of antibodies to CSP to promote direct binding of FcγRIII was significantly correlated with ADCC tested by the ADCC reporter cell line (r = 0.405, P = 0.022, Fig. 2F) and primary NK cells (r = 0.538, P = 0.002, Fig. 2H). FcγRs occur in populations as different alleles[40]. Therefore, we tested different alleles of FcγRIIa (R131, H131) and FcγRIII (F158, V158) for IgG-mediated binding[39]. Published data indicate that these alleles are prevalent in our study population[41]. Antibodies to CSP effectively promoted the binding of both alleles of FcγRIIa and FcγRIII. There was a high correlation of FcγR binding between the different alleles; rho = 0.8371 for FcγRIIa, and rho = 0.78 for FcγRIII (Fig. 3C, D).

**Regions of CSP targeted by functional antibodies**. Our earlier data indicated that CSP was a significant target of human antibodies that promote phagocytosis (Fig. 1F, G). Therefore, we next investigated different regions of CSP as targets of functional antibodies in malaria-exposed individuals using three constructs of CSP: (i) a synthetic peptide representing the NT region (which is not included in the RTS,S vaccine), (ii) a synthetic peptide representing the repeat region (NANP repeats), (iii) a recombinant protein representing the CT region. Each construct was separately coated onto fluorescent beads (at saturating concentrations) and opsonized with the pool of Kenyan sera. Antibodies effectively promoted neutrophil phagocytosis of beads coated with each construct (Fig. 4A). We subsequently assessed naturally acquired antibodies to regions of CSP for their ability to

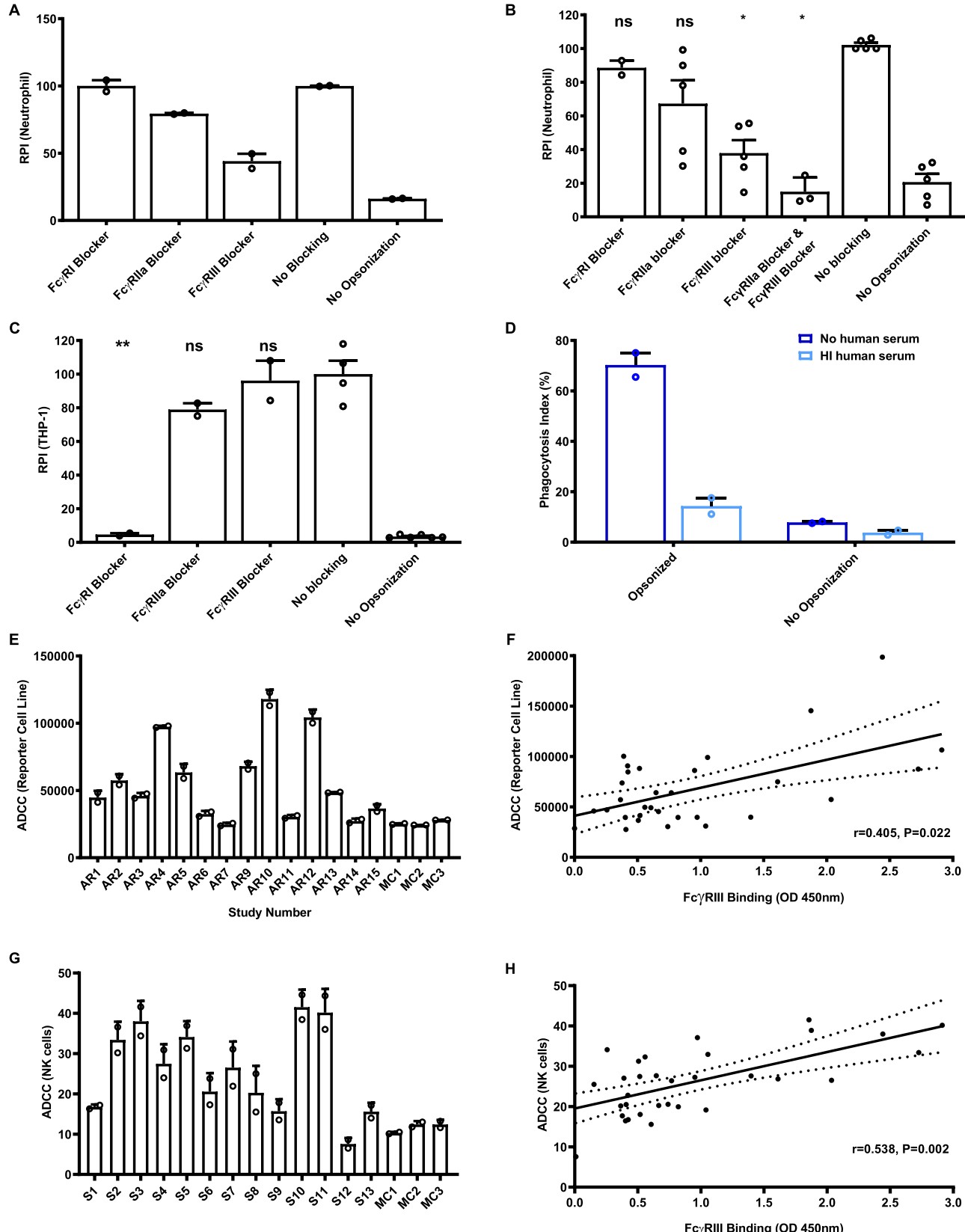

promote engagement of FcγRIIa and FcγRIII. Human antibodies bound each CSP region with sufficient density to promote binding of FcγRIIa and FcγRIII (Fig. 4B). As the total level of IgG binding varied among different CSP constructs (Supplementary Fig. 5), we standardised FcγRIIa and FcγRIII binding relative to

the level of IgG reactivity (termed FcγR-binding efficiency). Our data suggested that antibodies to the NT region had a higher potential to engage FcγRs than the central repeat and CT regions (activity was significantly higher for NT versus CT or NANP regions, $P < 0.001$ for each comparison).

**Fig. 2 Roles of specific Fcγ receptors in mediating opsonic phagocytosis of sporozoites. A** Effect of blocking-specific FcγRs on phagocytosis of sporozoites by neutrophils. *P. falciparum* sporozoites were opsonized with a pool of Kenyan adult sera. Bars represent mean and range from two experiments in duplicate (*P* < 0.001, one-way ANOVA). **B** Effect of blocking-specific FcγRs on phagocytosis of CSP-beads by neutrophils. CSP-coated beads were opsonized with a pool of serum from Kenyan adults. Bars represent mean and standard error from five experiments conducted in duplicate (*P* = 0.001, one-way ANOVA). **C** Effect of blocking FcγR receptors on phagocytosis of CSP beads by THP-1 cells. Bars represent mean and standard error from two experiments (*P* < 0.001 for FcγRI, one-way ANOVA). **D** Opsonic phagocytosis of CSP-coated beads by THP-1 cells was much lower in the presence of heat-inactivated human serum (light-blue bars) compared to the inclusion of FCS alone (dark-blue bars). Bars and error bars represent the mean and range from two independent experiments. **E** Antibodies from selected Kenyan serum samples induced ADCC activity using CSP-coated beads and an ADCC reporter cell line. Selected individuals (*n* = 13, AR1–AR13) are shown as examples to demonstrate the variation of activity observed (Kruskal–Wallis test *P* = 0.008). Sera from malaria-naive Melbourne donors were used as negative controls (MC), which were lower than the Kenyan serum samples (Mann–Whitney test *P* = 0.023). Bars and error bars represent the mean and range of the duplicates. ADCC values on the *Y* axis are luminescence units. **F** Correlation between antibody activity in FcγRIII binding and ADCC assays (reporter cell line) among selected Kenyan adult serum samples (*n* = 31) with high, intermediate or low total IgG against CSP (*r* = 0.405, *P* = 0.022, linear regression). The solid and dotted lines represent the linear regression line and 95% confidence intervals, respectively. **G** Antibodies from the Kenyan sera were tested for ADCC activity using CSP-coated beads and primary NK cells. Selected individuals (*n* = 13 S1–S13) are shown as examples of the variation of responses observed (Kruskal–Wallis test *P* = 0.015). Sera pool from malaria-naive Melbourne donors was used as negative controls (MC), which were lower than the Kenyan serum samples (Mann–Whitney test *P* = 0.03 for FcγRIIa and *P* = 0.05 for FcγRIII). Bars and error bars represent the mean and range of the duplicates. ADCC values on the *Y* axis represent the percentage of NK cells that are positive for CD107 staining by flow cytometry. **H** Correlation between antibody-mediated FcγRIII binding and ADCC activity (using primary NK cells) among selected Kenyan adult serum samples (*n* = 31) with high, intermediate or low total IgG against CSP (*r* = 0.538, *P* = 0.002, linear regression). The solid and dotted lines represent the linear regression line and 95% confidence intervals, respectively. * indicates significant difference (*P* < 0.05; ** indicates *P* < 0.01); ns not significant, RPI relative phagocytosis index.

We further evaluated the significance of different regions as targets of functional antibodies using antibodies generated against specific regions and whole sporozoites in phagocytosis assays. We first confirmed that rabbit anti-CSP antibodies can promote phagocytosis of sporozoites. *P. falciparum* sporozoites opsonized with these antibodies gave a high phagocytosis index (40.5%) compared to control (Fig. 4C). Subsequently, we established that a known mAb against the central NANP-repeat region of *P. falciparum* CSP (mAb 2H8, mouse IgG2a subclass)[33] promoted significant phagocytosis (17.4%, Fig. 4D) that was greater than seen with a negative control mAb 3D11 that targets CSP of *P. berghei* which does not have the NANP-repeat epitope (Fig. 4D). The NANP-repeat region encodes an immunodominant epitope, while much less is known about the significance of NT and CT regions as targets of antibodies. To further define the role of antibodies to the CT and NT regions, and complement data using human antibodies, we generated rabbit polyclonal antibodies to each region (Supplementary Fig. 6), and used these antibodies to opsonize transgenic PfCSP-*P. berghei* sporozoites for phagocytosis by neutrophils. Rabbit polyclonal antibodies to both the CT and NT regions effectively promoted phagocytosis of PfCSP-*P. berghei* sporozoites (Fig. 4E). Since the concentration of IgG to the CT and NT from vaccinated rabbit sera differed (Supplementary Fig. 6), we assessed the phagocytosis efficiency and antibodies, relative to IgG levels (calculated as phagocytosis activity divided by IgG reactivity determined by ELISA). This showed that antibodies to the N-terminal region had significantly higher phagocytosis efficiency compared to CT antibodies.

Since antibodies to the NT region were highly effective in activating FcγRs and promoting opsonic phagocytosis, we performed epitope mapping of the rabbit polyclonal antibodies against the NT region using an overlapping peptide array (Fig. 4F and Supplementary Fig. 6D). A peptide array was appropriate for epitope mapping because the NT region is predicted to be unstructured and contains no cysteines (PlasmoDB, https://plasmodb.org/plasmo/app/record/gene/PF3D7_0304600). Antibodies were reactive with peptides covering a 21-aa region (DDGNNEDNEKLRKPKHKKLKQ) that is close to the junction between the N-terminal region and the NANP/NVPD repeat region.

**Acquisition of antibodies that promote FcγR interactions and opsonic phagocytosis through natural exposure.** We analysed serum samples collected from a cohort of malaria-exposed young children and adults residing in western Kenya in a region with high malaria transmission intensity (from the Chulaimbo region, Supplementary Table 1). In this setting, the bulk of malaria morbidity and mortality occurs in children aged <5 years. In contrast, adults have substantial immunity, demonstrated by the lack of development of symptomatic malaria episodes, although asymptomatic parasitemia may occur[42]. We measured the ability of CSP-specific antibodies to interact with FcγRIIa and FcγRIII as well as promoting opsonic phagocytosis of CSP beads by neutrophils (Fig. 5A–D). The overall prevalence of antibodies with each functional activity was neutrophil phagocytosis 16%, (95% CI, 9.19, 26.38%), FcγRIIa binding 20% (95% CI, 12.28, 30.84%) and FcγRIII binding 14.7% (95% CI, 8.19, 24.85%). The level of IgG reactivity to CSP, FcγR binding and opsonic phagocytosis by neutrophils was low among young children and significantly increased with age. By contrast, adults with lifelong exposure had acquired significantly higher functional antibodies compared to children. We also evaluated antibodies that promoted phagocytosis of blood-stage merozoites, which had a substantially higher prevalence and level among young children compared to functional antibodies to CSP (Fig. 5B, E, F). This supports the concept that naturally acquired immunity to sporozoites is acquired much more slowly and requires substantially more exposure to develop, compared to blood-stage immunity. We investigated whether the higher functional activity among adults results only from the higher concentration of CSP-specific IgG or also reflects a higher functional potential of antibodies among adults. We analysed antibody functional activity relative to IgG reactivity against CSP; to do this, we excluded individuals with very low IgG to CSP (below the median OD of 0.140), and we calculated functional activity relative to IgG level for each individual. Interestingly, this demonstrated that the relative phagocytosis efficiency was significantly higher in adults compared to children (Fig. 5G) (Mann–Whitney test, *P* = 0.022). Further, there was a significant correlation between age and FcγR binding efficiencies (*r* = 0.354, *P* = 0.031 for FcγRIIa binding and *r* = 0.572 for FcγRIII binding, *P* < 0.001).

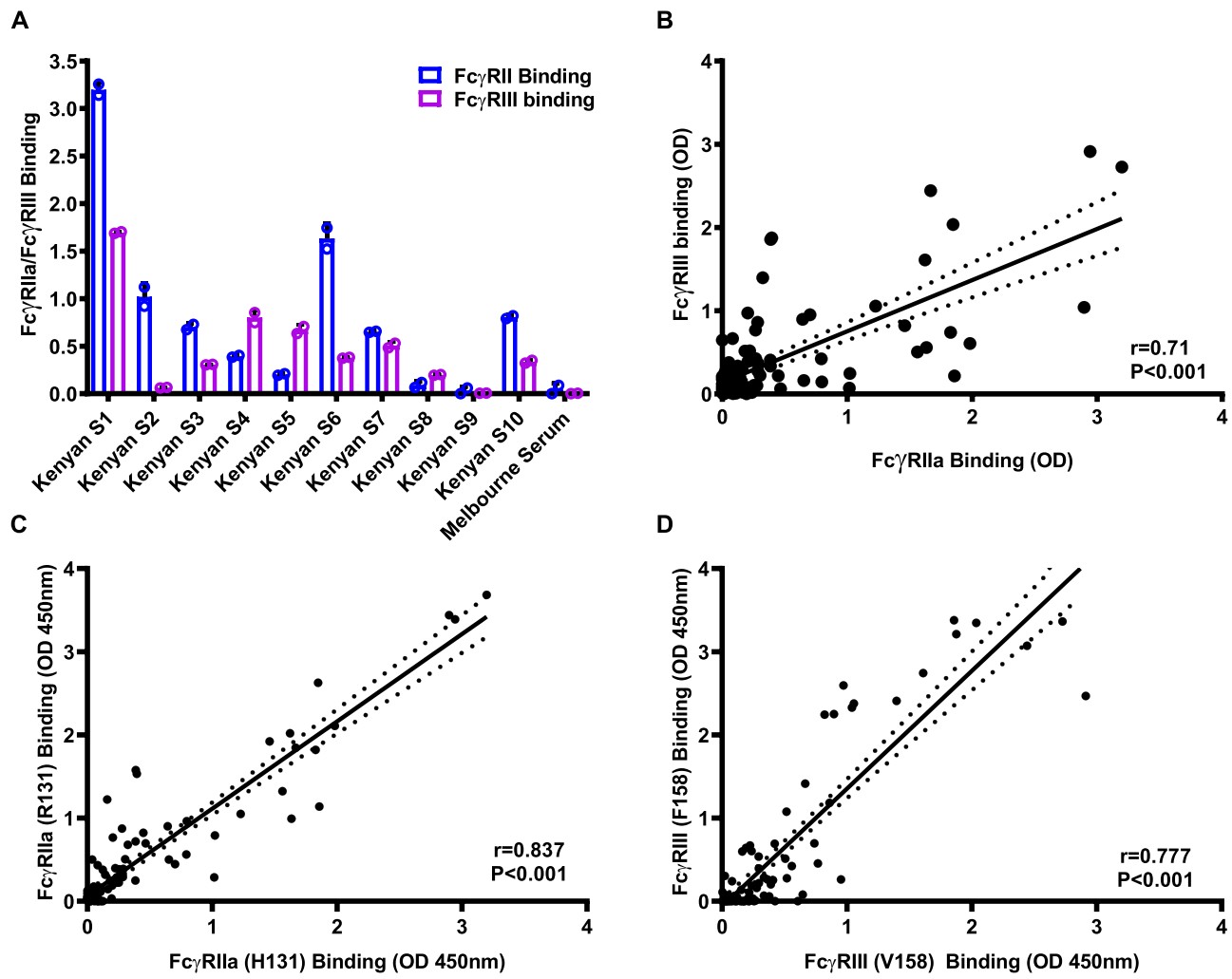

**Fig. 3 Direct quantification of FcγRIIa and FcγRIII binding activity. A** Antibodies to CSP among Kenyan adults were tested for their ability to promote binding to FcγRIIa and FcγRIII. Bars and error bars represent the mean and standard deviation for each sample tested in duplicate. Representative individuals ($n = 10$) are shown to reflect the range of activities observed. Sera from malaria-naive Melbourne donors were used as negative controls (MC), which were lower than the Kenyan serum samples (Mann–Whitney test $P = 0.026$). **B** Correlation between FcγRIIa binding and FcγRIII binding in the Kenyan adult cohort. There was a significant correlation between FcγRIIa binding and FcγRIII binding ($n = 104$, $r = 0.71$, $P < 0.001$, linear regression). The solid and dotted lines represent the linear regression line and 95% confidence intervals, respectively. **C** Correlation between antibody-mediated binding of two different alleles of FcγRIIa (R131 and H131) using samples from the Kenyan adult cohort ($n = 104$, Spearman's rho = 0.837, $P < 0.001$). The solid and dotted lines represent the linear regression line and 95% confidence intervals, respectively. **D** Correlation between antibody-mediated binding of two different alleles of FcγRIII (F158 and V158) using samples from the Kenyan adult cohort ($n = 104$, Spearman's rho = 0.777, $P < 0.001$). The solid and dotted lines represent the linear regression line and 95% confidence intervals, respectively.

**Opsonic phagocytosis by neutrophils correlated with FcγR binding and cytophilic IgG subclasses.** We extended our analysis to a larger cohort of immune adults in western Kenya (Kanyawegi cohort, $n = 104$) who have been exposed to life-long malaria in a region with high malaria transmission intensity. We reasoned that a large proportion of these adults are likely to have functional antibodies to CSP, facilitating analysis of antibody factors and specificities that determine functional opsonic phagocytosis activity. Indeed, many samples demonstrated substantial phagocytosis and FcγR binding activity. The prevalence of antibodies with each functional activity were neutrophil phagocytosis 47.1% (95% CI, 37.6, 56.9%), FcγRIIa binding 67.3%, (95% CI, 57.6, 75.8%) and FcγRIII binding 61.5% (95% CI, 51.7, 70.5%).

Opsonic phagocytosis by neutrophils was significantly correlated with FcγRIIa binding ($r = 0.50$, $P < 0.001$) and FcγRIII binding ($r = 0.43$, $P < 0.001$) (Fig. 6A). Multivariate regression analysis suggested that FcγRIII was the primary factor associated with

neutrophil opsonic phagocytosis (regression coefficient = 10.15, $P < 0.001$, Table 1), whereas FcγRIIa was less important. Previously we have reported that antibodies to CSP in this adult cohort are predominantly IgG1 and IgG3, with lower levels of IgG2, and very little IgG4[33]. IgG1 and IgG3 are the main subclasses that interact with FcγRIIa and FcγRIII to promote phagocytosis, and IgG2 has weak activity. Among Kenyan adults, neutrophil phagocytosis was significantly positively correlated with IgG1 ($r = 0.32$, $P < 0.001$), IgG2 ($r = 0.26$, $P = 0.007$) and more strongly with IgG3 ($r = 0.50$, $P < 0.001$) (Fig. 6B, C, Supplementary Fig. 7 and Supplementary Table 2). There was no correlation seen with IgG4 ($r = 0.09$, $P = 0.365$), though levels of IgG4 were low. In multivariate regression, IgG3 remained significantly associated with neutrophil phagocytosis, suggesting it is the major mediator of this activity (Supplementary Table 3). In addition, both FcγRIIa and FcγRIII binding were significantly correlated with total levels of IgG and IgG subclasses, primarily IgG1 and IgG3 (Supplementary Tables 2

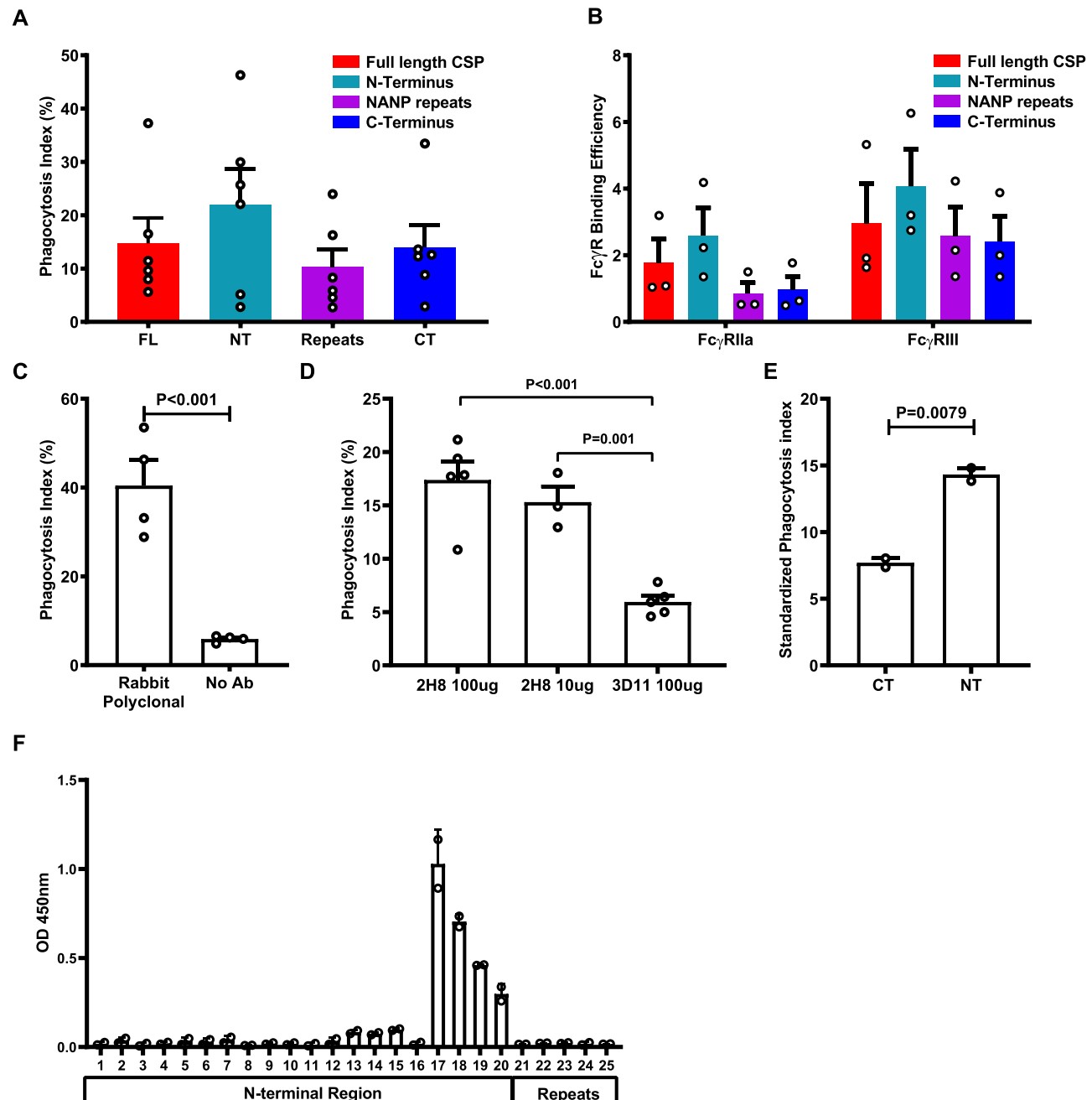

and 3). Overall, samples classified as high phagocytosis activity (defined as being above the median) had significantly higher levels of FcγRIIa and FcγRIII binding and IgG3, and had significantly higher levels of IgG to all regions of CSP, compared to samples with low phagocytosis activity (Fig. 6D, E).

Similar observations were found among the Chulaimbo cohort of children and adults. Opsonic phagocytosis significantly correlated with FcγR binding, and IgG1 and IgG3, with the correlation stronger for FcγRIII and IgG3 (Supplementary Tables 4 and 5). We applied these assays in a phase 1/2a clinical trial of the RTS,S vaccine. This showed that antibodies promoting opsonic phagocytosis by neutrophils and binding of FcγRIII and FcγRIIa were higher among protected than non-protected subjects[43].

## Discussion

We provide results defining mechanisms and targets of antibody-mediated immunity to sporozoites through interactions with

specific FcγRs expressed on immune cells. Our findings support a key role for neutrophils in the clearance of sporozoites in blood, mediated by FcγRIIa and FcγRIII interactions with IgG1 and IgG3 targeting the major sporozoite antigen, CSP, with additional contributions from monocytes (classical subset) and NK cells. Using intact sporozoites and antigen-coated beads, we established that phagocytosis, in whole blood, was predominantly by neutrophils, contrasting the low-level activity observed with monocytes. We defined key mechanisms involved by blocking specific FcγRs on cell surfaces and by using FcγR dimers as probes to bind antigen–antibody complexes. This identified the importance of FcγRIIa and FcγRIII, and data particularly point to the importance of FcγRIII. We defined antibody targets of this functional mechanism using antibodies to different regions of CSP, different CSP regions coated onto beads and chimeric *P. berghei* sporozoites expressing *P. falciparum* CSP. We established that CSP is a major target of these functional antibodies and all

**Fig. 4 CSP-specific antibodies induced opsonic phagocytosis by neutrophils. A** The level of opsonic phagocytosis by neutrophils of beads coated with different regions of CSP. Beads coated with full-length CSP (FL), the CT region, the NT region and the NANP repeat region (repeats) were opsonized with antibodies from a pool of Kenyan adult donors. Unopsonized beads were used as a negative control for assay background. Data show mean and standard error from six experiments ($P = 0.051$ and $P = 0.187$ for comparisons between NT and NANP or CT domains, respectively; one-way ANOVA). The level of phagocytosis activity was expressed relative to the IgG reactivity to each region (see Supplementary Fig. 8A). **B** Antibodies from the Kenyan adult pool were tested for their ability to promote the binding of FcγRIIa and FcγRIII binding activity with each region of CSP. The level of FcγR binding was standardised according to the level of total IgG binding to each CSP construct (FcR-binding efficiency; see Supplementary Fig. 8A). Data show mean and standard error based on three experiments. FcγRIIa and FcγRIII binding to the NT region was significantly higher compared to the NANP-repeat region or CT region ($P < 0.001$, two-way ANOVA). **C** Opsonic phagocytosis of *P. falciparum* sporozoites by neutrophils mediated by rabbit polyclonal antibody to CSP, compared to no-antibody (no Ab) control ($P < 0.001$, unpaired *t* test). Data show mean and standard errors from four experiments. **D** Opsonic phagocytosis of *P. falciparum* sporozoites by neutrophils mediated by mouse monoclonal antibody (2H8) against CSP at different concentration, compared to mouse monoclonal antibody (3D11) against *P. berghei* CSP as a control ($P = 0.0025$, one-way ANOVA). Data shown are mean and standard errors from four experiments. **E** Rabbit IgG to the NT region was more efficient in promoting phagocytosis of PfCSP-*P. berghei* sporozoites by neutrophil compared to rabbit IgG to the CT region ($P = 0.0079$, unpaired *t* test). Data shown are mean and standard error based on two experiments. Phagocytosis activity was standardised relative to the level of IgG to CSP of each rabbit antibody. **F** Rabbit antibodies generated against the NT region were tested for reactivity against an array of overlapping peptides (15 aa with 12 aa overlap) including the NT region, NANP/NVDP repeat and the junctional region. Data show IgG reactivity to different peptides; the reactive peptides include a 21-aa region corresponding to CSP residues 76–96 ($P = 0.036$ for differences in peptide reactivity, Kruskal–Wallis test, two experiments). Error bars represent the range from two experimental repeats. Peptide sequences are provided in Supplementary Fig. 8B.

three regions of CSP can be targeted by antibodies to engage FcγR and promote phagocytosis. In particular, antibodies to the NT region had significant activity across a range of different assays, providing a potential lead for vaccine development since the NT region is not included in the CSP-based vaccines RTS,S (now in implementation studies) or R21 (in phase 2 trials). Further, we established that naturally acquired antibodies were capable of promoting opsonic phagocytosis of sporozoites. The acquisition of functional antibodies was age-dependent and higher levels of opsonic phagocytosis activity and binding of FcγRIIa and FcγRIII were only observed with samples from adults but not young children. Functional activity correlated with IgG1 and IgG3 subclasses.

The higher phagocytosis activity of neutrophils, but the very low-level activity of monocytes, may relate to the presence and use of different FcγRs; neutrophils had a greater rate of phagocytosis, which was not simply explained by the higher abundance of neutrophils compared to monocytes. Interestingly, blocking either FcγRIIa or FcγRIIIa/b inhibited phagocytosis activity by neutrophils, suggesting that both receptors are required for phagocytosis, which was supported by achieving greater inhibition with blocking antibodies to FcγRIIa and FcγRIII used in combination. Blocking FcγRI had no effect on phagocytosis by neutrophils. Several observations suggested the higher importance of FcγRIII; there was generally greater inhibition of sporozoite phagocytosis by FcγRIII blockade, and FcγRIII binding by antibodies was more highly correlated with phagocytosis than FcγRIIa in multivariate analysis.

FcγRIIIb, which is GPI-anchored, is the most abundant FcγRIII type expressed on neutrophils, but recent studies have reported that neutrophils also express FcγRIIIa[30], which has a transmembrane and cytoplasmic domain for intracellular signalling. For phagocytosis, intracellular signalling may occur via FcγRIIIa or FcγRIIa, with binding to FcγRIIIb also being important, as indicated by our findings. If FcγRIII and FcγRIIa act cooperatively in phagocytosis, this may explain why there was the limited activity of FcγRIIa in phagocytosis by monocytes. NK cells express FcγRIIIa, and we established that antibodies to CSP can promote NK activation through interaction with FcγRIIIa. This may lead to the killing of sporozoites as well as the release of pro-inflammatory cytokines[44]. However, NK cells comprise only 1–5% of leukocytes in blood suggesting they would play a less prominent role in immunity against sporozoites than neutrophils, which comprise 50–70% of leukocytes. Sporozoites are also be

exposed to dermal macrophages and liver Kupffer cells, which could involve additional mechanisms of FcγR-mediated clearance and warrant investigation in future studies, but the relative frequency of these cells and their expression of FcγRs is different to phagocytes in blood.

Existing studies on antibody-mediated opsonic phagocytosis in malaria have largely used the THP-1 cell line as a model, using standard conditions with FCS (but not including human serum). Under these conditions, we found that phagocytosis was effectively inhibited by blockade of FcγRI, but minimal inhibition was seen with blocking FcγRIIa or FcγRIII. A recent study reported that opsonic phagocytosis by THP-1 cells was not correlated with protection in a RTS,S phase I/IIa trial in malaria naïve adults[45]. We showed that non-specific IgG in human serum can also reduce THP-1 cell phagocytosis activity because FcγRI interactions are inhibited by monomeric IgG presented in human plasma, whereas monomeric bovine IgG does not bind human FcγRI. Together, these findings suggest THP-1 cells are not an ideal model for studying opsonic phagocytosis against sporozoites.

We established that all three regions of CSP can be targeted by antibodies that engage FcγRIIa and FcγRIII to promote phagocytosis of sporozoites by neutrophils, thereby ascribing functions for antibodies to these regions that have not been previously reported. Of note, there was the significant activity of antibodies to the NT region as seen in phagocytosis assays using human antibodies with antigens-coated beads, phagocytosis of sporozoites by rabbit antibodies raised to different regions, and FcγR binding by human antibodies. These findings suggest a greater focus on the NT region in vaccine design may be valuable, in addition to the inclusion of the central repeat and CT regions. Current leading vaccines do not include the NT region[3,46]. We mapped antibody epitopes to a 21-aa region that is close to the junction between the NT and central repeat regions, but does not include the recently described junctional epitope[34] and is distinct from the Region I epitope, which is a site for N-terminal cleavage at the time of hepatocyte invasion[47,48].

Our observations that antibodies acquired through exposure to malaria that can promote FcγR-mediated phagocytosis by neutrophils further support the concept that this is a mechanism that occurs in vivo among individuals who develop high levels of immunity. In a setting of high malaria transmission in Kenya, these functional antibodies are very low among young children who are most at risk of infection and severe malaria but occur at

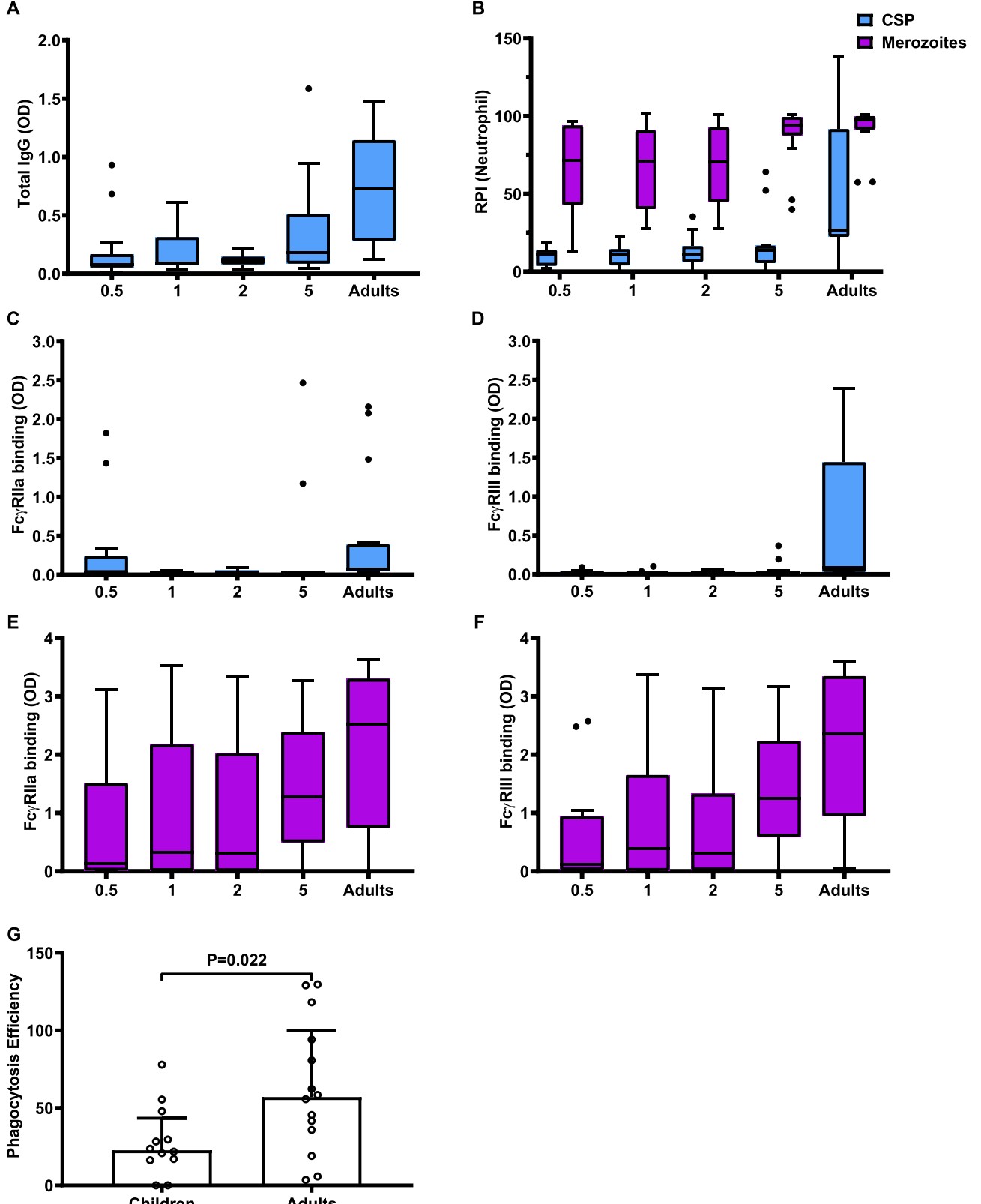

much higher levels among some adults with lifelong exposure. Notably, functional antibodies to CSP were acquired much more slowly and required greater exposure to develop, compared to antibodies that promote the phagocytosis of blood-stage merozoites. Using FcγR dimers as probes, we found that acquired antibodies to CSP could promote interactions with FcγRIIa and

FcγRIII, and there was similar binding to two different alleles of each receptor. Of further interest, we found that the relative functional activity of antibodies in promoting neutrophil phagocytosis was higher among adults than children; currently, there is limited knowledge on the differences in functional activities of antibodies to sporozoites among children of different ages and

**Fig. 5 Acquisition of antibodies that promote neutrophil opsonic phagocytosis and FcγR binding among Kenyan children and adults.** Samples from Kenyan children and adults ($n = 75$) were tested for total IgG binding to CSP (**A**), opsonic phagocytosis of CSP beads (blue bar) or whole merozoites (purple bars) by neutrophils (**B**), FcγRIIa binding (**C**) and FcγRIII binding (**D**) against CSP (blue bars) and FcγRIIa binding (**E**) and FcγRIII binding (**F**) against merozoites (purple bars). Individuals were categorised into age groups of 0–1 (0.5), 1–2 (1), 2–4(2), 4–18 (5) 18 or more years (adults). The total level of IgG to CSP was significantly lower among children compared to adults ($P < 0.001$, Kruskal–Wallis Test). Opsonic phagocytosis of merozoites and CSP-coated beads was also lower among children compared to adults ($P < 0.001$, respectively, Kruskal–Wallis Test), with a higher level of opsonic phagocytosis against merozoites compared to CSP-coated beads ($P < 0.001$, Kruskal–Wallis Test). Similar trends were seen for FcγRIIa and FcγRIII binding to CSP ($P < 0.001$, respectively, Kruskal–Wallis Test). The level of FcγRIIa and FcγRIII binding to merozoites was also varied among age groups ($P = 0.018$ for FcγRIIa and $P = 0.003$ for FcγRIII, respectively, Kruskal–Wallis Test). **A**–**F** Boxes and whiskers indicate the medians and interquartile range (IQR) and 1.5 × IQR of each group. Values exceeding this range are presented as dots. **G** Opsonic phagocytosis activity by neutrophils was expressed relative to IgG reactivity against CSP (phagocytosis efficiency) for each individual. Phagocytosis efficiency was significantly higher for adults compared to children (Mann–Whitney U test, $P = 0.022$). Bars and error bars represent the median and interquartile range for each group.

adults. Prior studies in this population found that high levels of antibodies to CSP among adults were associated with a reduced risk of infection among adults[49]. In contrast, a cohort of predominantly children found no association in Mali, where malaria transmission intensity was lower than our study population[50]. Interestingly, the relative levels of binding activity with each FcγR varied significantly among individuals, which is likely to impact on phagocytosis activity. Variations in specific FcγR binding will be influenced by IgG subclass profiles, epitopes targeted by antibodies and IgG glycosylation[22]. IgG1 and IgG3 predominated among acquired IgG subclasses and correlated with FcγR-binding activity.

In this study, we adapted the use of antigen-coated microspheres to study opsonic phagocytosis activity of antibodies to CSP, the key immune target of *P. falciparum* sporozoites. An advantage of this approach is that it overcomes the challenges in generating large quantities of sporozoites for immunologic assays, which is difficult to achieve for large population studies and clinical trials. The level of opsonic phagocytosis measured using CSP-coated beads was highly correlated with that measured using intact sporozoites. This platform also enabled us to dissect the functional antibody responses to each region of CSP. We also confirmed phagocytosis using cryopreserved sporozoites (viability > 85%[51]), and freshly dissected sporozoites.

Prior studies have demonstrated that antibodies to sporozoites can inhibit sporozoite motility, cell traversal and hepatocyte invasion as potential immune mechanisms. However, there is currently no consistent evidence showing these mechanisms correlate with protection against *Plasmodium* infection in clinical studies[3]. In passive immunisation models in animals, high levels of IgG to CSP can protect against experimental infection[3,4]. However, similar high levels of protection, sustained over time, have not yet been achieved by vaccine-induced antibodies in clinical trials[52–56]. RTS,S induces very high CSP IgG levels, but only provides modest protection[53,56], suggesting that other strategies are needed to generate antibodies that have higher functional protective activity[3,4]. Our data provide insights into a largely understudied immune mechanism of antibodies in the phagocytic clearance of sporozoites and suggest that harnessing FcγR-mediated mechanisms could be a strategy to generate more potent immunity for higher levels of protection. This could be achieved by vaccine strategies that optimise IgG subclass responses, reduce IgM that may inhibit FcγR interactions and improve targeting of specific functional epitopes. These advancements could be achieved through the use of specific vaccine adjuvants and immunogen design. FcγR interactions with IgG bound to antigens can also be influenced by epitope density and the spatial orientation and positioning of epitopes[57,58]. Structural biology approaches may be valuable to understand this and inform vaccine design. It is also possible that different regions of CSP mediate immunity through different immune mechanisms.

This has been shown for influenza hemagglutinin, with FcγR-mediated mechanisms being important for antibodies to the conserved stalk region, but less important for strain-specific antibodies to the head region[18,21].

Future studies investigating the induction of these functional antibodies and their correlation with protection are warranted in phase II and phase III trials of RTS,S, and other CSP-based vaccines. We have conducted an initial study in a phase I/IIa trial of RTS,S[56]. This established that RTS,S vaccination does indeed induce antibodies to CSP that promote binding of FcγRIIa and FcγRIII, as well as phagocytosis by neutrophils. Importantly, these functional activities were significantly higher among subjects who were protected against malaria infection versus those who developed infection[43]. While CSP is a target of acquired human immunity, including functional antibodies, there is currently limited understanding of potential differences in naturally acquired and vaccine-induced immunity against sporozoites. Further research on these differences may reveal important insights to inform vaccine development.

In summary, our studies have defined key mechanisms and targets of FcγR-mediated immunity against sporozoites in blood. Our findings point to a major role for neutrophils in the phagocytic clearance of sporozoites, defined important roles for FcγRIIa and FcγRIII and established CSP as a key target for this functional activity. Monocytes also contribute to phagocytosis but have lower activity, and antibodies can also promote NK cell ADCC activity through the engagement of FcγRIIIa. Understanding and defining mechanisms that mediate immunity is crucial to enable the development of more efficacious vaccines. Future studies should investigate how effectively current vaccines in clinical trials generate antibodies with these functional activities. Harnessing this knowledge in vaccine development may be an effective strategy for generating a potent immune response that provides higher levels of protection against malaria.

## Methods

**Study populations and ethics statement.** Plasma from malaria-exposed healthy, asymptomatic adults living in the Kanyawegi sub-district of Kisumu County, Kenya ($n = 104$, age range 18–79; 26.9% male) was collected in August 2007 (Kanyawegi cohort)[59]; no participants had symptoms of malaria and 46.15% was asymptomatic *P. falciparum* parasitemia (detected by microscopy). Plasma from healthy, asymptomatic young children and adults living in the Chulaimbo sub-district of Kisumu County, Kenya ($n = 75$, age range 0.3–5.9 years for children, age range 19.6–69.2 years for adults; 55% female) was collected in February–March 2007 (Chulaimbo cohort)[42]; cohort details are summarised in Supplementary Table 1. Transmission intensity was relatively high at the time blood was collected (prevalence of asymptomatic parasitemia in children ≤10 years was 70–80%)[60]. Blood was collected from healthy male and female adults for use in whole leukocyte, neutrophil, monocyte and NK assays.

Ethics approval was obtained from the Alfred Hospital Human Research and Ethics Committee (protocol 385-18), the Institutional Review Board for Human Investigation at University Hospitals of Cleveland for Case Western Reserve University, USA (protocol 02-04-04) and the Ethical Review Committee at the

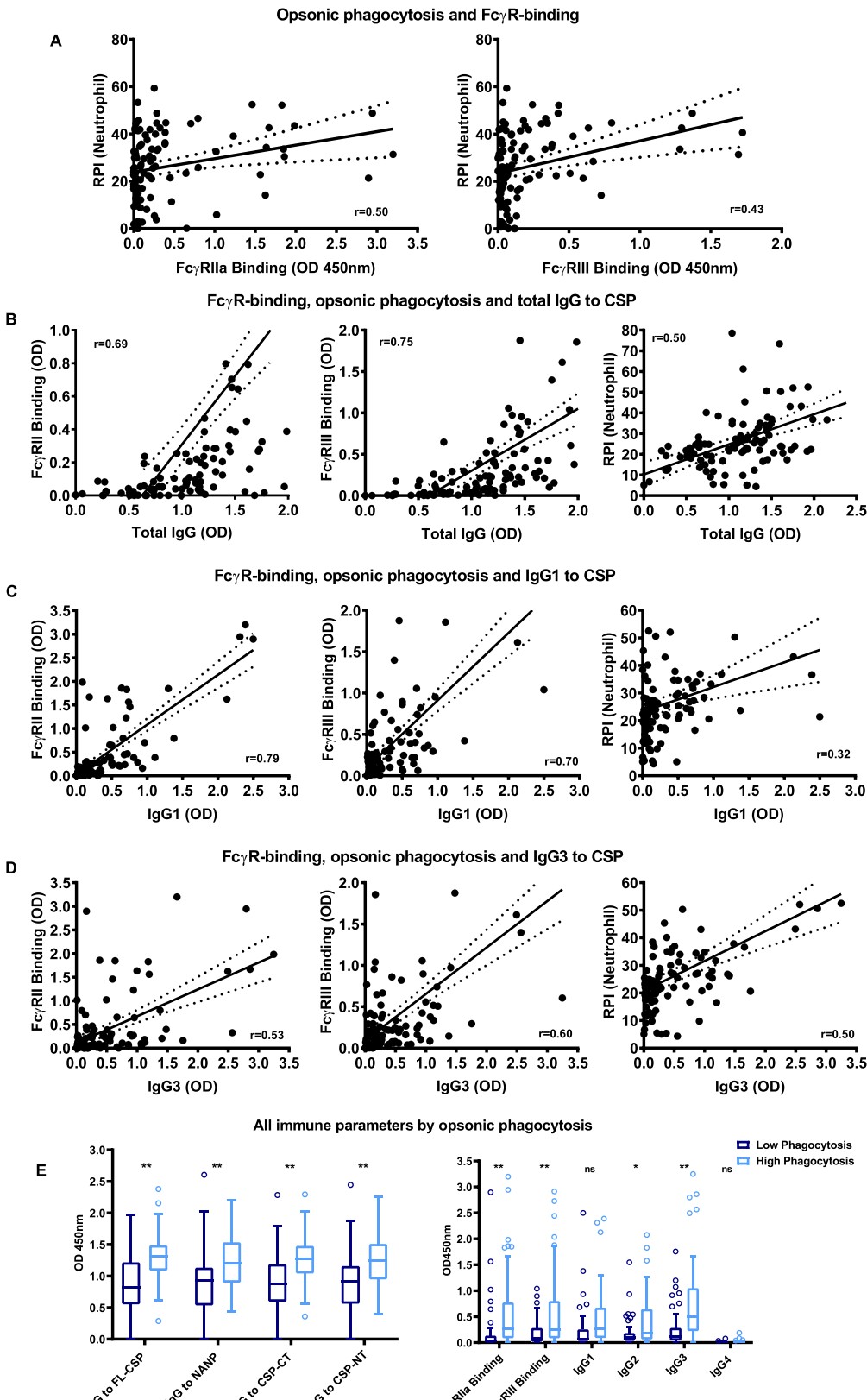

Kenya Medical Research Institute, Kenya (protocol SSC867). Written informed consent was obtained from all study participants or their parents or legal guardians.

**Antigens**. The antigens were used in this study were all based on the 3D7 reference sequence of *P. falciparum* CSP (Gene ID PF3D7_0304600; www.plasmodb.org). The recombinant full-length CSP was purchased from Sanaria (Sanaria, Rockville, USA).

The protein was expressed in *Pichia pastoris* beginning at amino acid residue 50 and contains $(NANP)_{22}$ repeats and $(NVPD)_4$ repeats. The C-terminal region of CSP (CT) and a fusion of the N-terminal region and C-terminal region of CSP (N + C) were generated as recombinant proteins using HEK293F cells (purchased from Invitrogen, Australia). The N-terminal region of CSP (amino acid 19-104, NT) was generated as synthetic peptides and the central repeat region of CSP (NANP) were also generated as synthetic peptides (NANP×15) by LifeTein (LifeTein, NJ, USA).

**Fig. 6 Correlations between neutrophil opsonic phagocytosis, FcγR-binding and CSP-specific IgG subclasses in the Kenyan adult cohort. A** Among Kenyan samples ($n = 104$), levels of opsonic phagocytosis were significantly correlated with FcγRIIa binding and FcγRIII binding. **B–D** Antibody-mediated FcγRIIa binding, FcγRIII binding and opsonic phagocytosis were positively correlated with IgG (**B**), IgG1 (**C**) and IgG3 (**D**) reactivity to CSP among samples. All correlations were statistically significant ($P < 0.001$, Spearman's correlation). **E** Samples ($n = 104$) with higher opsonic phagocytosis activity (defined as greater than the median) showed higher levels of total IgG to full-length CSP ($P < 0.001$, two-way ANOVA), the NANP repeat region ($P = 0.001$, two-way ANOVA), the C-terminal region ($P < 0.001$, two-way ANOVA) and the N-terminal region ($P < 0.001$, two-way ANOVA) compared to samples with lower opsonic phagocytosis. FcγRIIa and FcγRIII binding by antibodies to CSP was also significantly higher among individuals with high opsonic phagocytosis ($P < 0.001$ and $P < 0.001$ respectively, two-way ANOVA). The levels of CSP-specific IgG2 and IgG3 against were significantly higher in samples with high opsonic phagocytosis ($P = 0.0464$ and $P < 0.001$ respectively, two-way ANOVA), but not clearly higher for IgG1 ($P = 0.106$, two-way ANOVA) and IgG4 ($P > 0.999$, two-way ANOVA). The solid and dotted lines represent the linear regression line and 95% confidence intervals. Boxes and whiskers indicate the medians and interquartile range (IQR) and $1.5 \times$ IQR of each group. Values exceeding this range are presented as dots. Asterisks indicate statistically significant levels (*$P < 0.05$, **$P < 0.01$).

**Table 1 Multivariate analysis of the association of opsonic phagocytosis of CSP-coated beads by neutrophils with FcγRIIa, FcγRIII, IgG subclasses and in the Kenyan adult cohort.**

|  | Regression coefficient* | 95% CI | P value |
|---|---|---|---|
| *Kanyawegi cohort* |  |  |  |
| FcγRIIa binding | 2.41 | −2.71, 7.53 | 0.353 |
| FcγRIII binding | 10.15 | 4.24, 16.05 | <0.001 |
| *Kanyawegi cohort* |  |  |  |
| IgG1 | 4.44 | −1.06, 9.94 | 0.113 |
| IgG2 | −1.38 | −6.45, 3.69 | 0.590 |
| IgG3 | 9.77 | 5.78, 13.76 | <0.001 |
| IgG4 | 3.94 | −109.25, 117.14 | 0.945 |

Kanyawegi cohort, $N = 104$ participants. *β values: linear regression coefficient.

A more detailed description of expressing recombinant CT and N + C in HEK293F cells can be found in Supplementary Information—Supplementary Methods.

**Generation of antibodies to different CSP proteins**. Rabbit vaccinations were performed by the Walter Eliza Hall Institute antibody facility; approval was provided by the Animal Ethics Committee of the Walter and Eliza Hall Institute (ref: 2020.019, 2014.009). Briefly, rabbits ($n = 2$) were vaccinated with one of the following antigens: full-length CSP, N + C recombinant protein and NT peptide. Each rabbit received three doses of antigen (200 μg/dose) co-administered with Freud's adjuvant on day 0, day 28 and day 56. Rabbit sera were collected after the third vaccination on day 68, and rabbit IgG was purified by protein A-positive selection. The rabbit anti-N + C IgG were initially tested, and shown to predominantly recognise the C-terminal region of CSP, and not the N-terminal region (Supplementary Fig. 6A). The rabbit anti-NT IgG was confirmed to recognise the N-terminal region of CSP (Supplementary Fig. 6B). Both NT and CT antibodies reacted with full-length CSP (Supplementary Fig. 6C). Monoclonal antibody (mAb) MGG4[34], which was isolated from an individual after repeated infection with attenuated sporozoites, was expressed and purified as a recombinant IgG1 antibody in HEK293 cells using established approaches. The mAb was tested against recombinant CSP to confirm specificity. Anti-CSP mAb 2A10[61] was expressed with LALA mutations in the Fc region[62] to inhibit FcγR interactions as a control antibody.

**Isolation of PBMC, neutrophils and whole leukocytes from peripheral blood**. Peripheral blood mononuclear cell (PBMC) and neutrophils were isolated from peripheral blood using previously established methods[63,64]. Briefly, peripheral blood from healthy donors was separated using Ficoll gradient centrifugation. The Buffy layers, which contain PBMCs and the RBC pellet that contains neutrophils, were separately collected. For PBMC isolation, cells collected from the Buffy layer were washed four times with PBS-1% NCS at 4 °C. PBMCs were subsequently resuspended in RPMI1640-10% foetal bovine serum (FCS) and kept on ice. Neutrophils were enriched by dextran segmentation from the RBC pellet post Ficoll gradient centrifugation, followed by hypotonic lysis[63]. Whole leukocytes were isolated from whole blood by initial enrichment with dextran segmentation followed by hypotonic lysis. Isolated neutrophils and whole leukocytes were adjusted to $5 \times 10^5$ neutrophil/ml or $5 \times 10^6$ phagocytes (monocytes and neutrophils)/ml and subsequently resuspended in RPMI1640 supplemented with 10% FCS and 2.5% heat-inactivated human serum and kept on ice. Cells were phenotyped for FcγR expression using antibodies to FcγRI (Clone 10.1 BD Bioscience, 1:50),

FcγRIIa (Clone 8.26, BD Bioscience, 1:25) and FcγRIII (Clone 3G8 BD Bioscience, 1:50). Antibodies to CD14 (Clone M5E2, BD Bioscience, 1:20) was also used to determine monocyte subsets.

**Maintenance of THP-1 cells**. Cells of the THP-1 promonocytic cell line (sourced from ATCC) were maintained in RPMI-1640 with 0.002 mol/L L-glutamine, 0.01 mol/L HEPES and 10% foetal calf serum (FCS). Cell density was monitored closely and maintained between $1 \times 10^5$ and $1 \times 10^6$ cell/ml. Cells were passaged every 6 days or when cell density approached $1 \times 10^6$ cells/ml.

**Culture and isolation of *P. berghei* sporozoites**. *Anopheles stephensi* mosquitoes were fed on Swiss Webster mice infected with a chimeric CSP expressing *P. berghei* ANKA parasite line[12]. Salivary glands were dissected 18–22 days post the infectious blood meal and homogenised in PBS. The sporozoites were passed through a 70-μm tissue strainer (Falcon) and counted using a haemocytometer.

**Culture and isolation of *P. falciparum* merozoites**. The D10 line of *P. falciparum* was maintained in RPMI-HEPES supplemented with 0.5% Albumax (Life Technologies) and 0.18% NaHCO₃. Cultures were maintained below 10% parasitemia and synchronised by sorbitol treatment. The merozoites were isolated using previously published methods[32]. Briefly, the schizoint stage of the infected erythrocytes was enriched by magnetic purification and continued cultured in a medium supplemented with transepoxysuccinyl-L-leucylamido(4-guanidino)butane (E64, Sigma-Aldrich) for 8 h before passing the mature schizoints through a 1.2-μm filter to release the merozoites. The merozoites were adjusted to $5 \times 10^7$/ml for opsonic phagocytosis assay or $1 \times 10^7$/ml for the coating to an ELISA plate for FcγR-binding assay.

**Covalent coupling of antigens to fluorescent latex beads**. Amine-modified fluorescent latex beads of 2.0-μm size (Sigma-Aldrich) were washed twice with 400 μl of PBS and centrifuged at $2000 \times g$ for 3 min. Glutaraldehyde of 8% in PBS was added to the beads and incubated on a roller overnight at 4 °C. After washing with PBS, 1 mg/ml of recombinant CSP was added to the beads and incubated on a vortex for 6 h. Subsequently, the beads were centrifuged and resuspended in 200 μl of ethanolamine and incubated for 30 min on a vortex to quench all the remaining amine groups. The beads were subsequently washed in PBS and blocked with 1% BSA overnight at 4 °C. The antigen-coated beads were kept in a sonicating water bath for 20 min at 4 °C to reduce aggregation and subsequently adjusted to $5 \times 10^7$ beads/ml. The beads were stored at 4 °C in the presence of 0.1% SDS and 0.02% NaN₃.

**Opsonic phagocytosis assays**. Opsonic phagocytosis assays were performed in variable settings for different purposes as described below and in Supplementary Table 6.

For opsonic phagocytosis of *P. falciparum* and *P. berghei* sporozoites using isolated cells, cryopreserved *P. falciparum* sporozoites were provided by PATH-MVI (Sanaria, Rockville, USA) and stored in liquid nitrogen, and were thawed at 37 °C for 40 s as described in the manufacturer's user manual. Freshly dissected *P. berghei* sporozoites that had been transfected to express the PfCSP[12] were also used in this study. The same approach was used for *P. falciparum* and *P. berghei* sporozoites in the experiments. The sporozoites were stained with 10 μg/ml ethidium bromide for 1 h on ice followed by three washes with RPMI-HEPES. For opsonization, 50,000 sporozoites were incubated with 1 μl of test serum for 1 h on ice, followed by co-incubation with 5000 neutrophils at a concentration of $5 \times 10^5$ cells/ml in RPMI-1640 supplemented with 10% FCS and 2.5% heat-inactivated human serum from malaria-naive Melbourne donors. An additional 1ul of test serum were also added to account for the volume change from adding neutrophils and maintain the same test antibody concentration during phagocytosis. The co-incubation was allowed for 15 min at 37 °C in a 5% CO₂ incubator for phagocytosis to occur. Cells were centrifuged at 4 °C and washed with cold PBS-1%NCS–0.04% NaN₃ (FACS buffer) to stop phagocytosis activity. The level of phagocytosis was quantified by flow cytometry (FACS CantoII, BD Biosciences) and analysed using

FlowJo software (version 10). The data were presented as phagocytosis index (PI), which was defined as the proportion of neutrophils (Supplementary Fig. 4A) or THP-1 cells (Supplementary Fig. 1A) that had phagocytosed sporozoites. For experiments on the Kanyawegi and Chulaimbo cohort, PI was further standardised as relative phagocytosis index (RPI), which was defined as the percentage of PI for each sample in relative to the positive control. Unopsonized sporozoites were included as negative controls in all assays. Standard neutrophil phagocytosis assays included 2.5% human serum from malaria non-exposed donors. Standard THP-1 phagocytosis assays were performed using established methods that include 10% FCS.

For phagocytosis of antigen-coated beads or merozoites using isolated cells, the phagocytosis assay was adapted from previously published methods with modifications[32]. Briefly, $1 \times 10^6$ antigen-coated fluorescent latex beads or merozoites were opsonized with serum samples for 1 h. The beads or merozoites were washed thrice with RPMI-1640 before co-incubation with $1 \times 10^5$ neutrophils for phagocytosis. Phagocytosis was allowed to occur for 20 min for beads or 10 min for merozoites at 37 °C and the cells were subsequently washed with FACS buffer at $300 \times g$ and 4 °C for 4 min. The proportion of neutrophils containing fluorescent beads or merozoites was evaluated by flow cytometry (FACS CantoII, BD Biosciences) and analysed using FlowJo software (Supplementary Fig. 4A). In some assays, the THP-1 cell line was used with standard conditions that include 10% FCS; in other assays, an additional 2.5% human serum from malaria naive Melbourne donors was used for comparison.

For phagocytosis of antigen-coated beads or *P. berghei* sporozoites using whole-blood leukocyte preparations, the CSP-coated beads were opsonized with a rabbit polyclonal antibody raised against CSP. Phagocytosis was performed by co-incubating the opsonized beads with the total leukocyte fraction from whole blood containing $1 \times 10^6$ phagocytes (monocytes and neutrophils) for 20 min at 37 °C in a 5% $CO_2$ incubator. These experiments were performed using variable antibody concentrations for opsonization or variable bead to phagocyte ratios. For testing opsonic phagocytosis of *P. berghei* sporozoites by whole-leukocyte preparations, $5 \times 10^5$ sporozoites were opsonized with a pool of malaria exposed Kenyan adult sera at a dilution of 1:100, followed by co-incubation with whole leukocytes containing $5 \times 10^4$ phagocytes. The cells were centrifuged at 4 °C for 4 min to stop the phagocytic activity and different phagocytes (monocytes and neutrophils) were labelled with fluoresce-conjugated antibodies for further analysis. Monocytes were labelled with anti-CD14-Alexa647 (MφP9 BD Bioscience, 1:33) and anti-CD16-BV421 (3G8 BD Bioscience, 1:20) and neutrophils were labelled with anti-CD66b-APC-H7 (G10F5 BD Bioscience, 1:33). For FACS analysis, phagocytosed beads were selected as PE^high and FSC^high population. Beads that were phagocytosed by monocytes and neutrophils were further distinguished using CD14 and CD66b staining. The number of beads phagocytosed by monocytes and neutrophils was calculated based on the fluorescence intensity (Supplementary Fig. 8). This number was further standardised using the total number of phagocytes (monocytes and neutrophils) and expressed as beads per 100 phagocytes, representing the phagocytosis rate of different cell types.

For inhibiting opsonic phagocytosis by blocking Fcγ receptors or depleting CSP antibodies, different FcγRs were blocked using specific blocking antibodies. FcγRI was blocked with 50 μg/ml blocking antibody (mAb 10.1 Merck), FcγRIIa was blocked with 50 μg/ml blocking antibody (mAb IV.3), FcγRIII was blocked with 50 μg/ml blocking antibody (mAb 3G8). THP-1 cells and neutrophils were treated with each FcγR blocker, respectively, at 4 °C for 30 min before co-incubation with CSP-coated latex beads, which have been opsonized with serum from immune adults. Phagocytosis assay was performed as mentioned above. To assess the role of antibodies to CSP in promoting phagocytosis mediated by acquired human antibodies, a pool of Kenyan adult serum was pre-incubated for 30 min with recombinant CSP and NT region peptide at a concentration of 20 μg/ml to bind CSP-specific antibodies, prior to use in phagocytosis assays.

**Measuring total IgG and IgG subclasses by ELISA.** Total IgG and IgG subclasses to recombinant proteins were measured by ELISA as previously described[33]. Briefly, CSP was coated at 1 μg/ml on 96-well Maxisorp microtiter plates (NUNC, Roskilde, Denmark) and incubated overnight at 4 °C. Plates were washed with PBS containing 0.05% Tween20 (PBS-Tween) and blocked with PBS-1% BSA at 37 °C for 2 h. Human serum or rabbit IgG were diluted to the desired concentration and added to the plates in duplicates. For total IgG detection, HRP-conjugated goat-anti-human IgG was used at 1:2500 dilution followed by ABTS substrate for colour development. The reaction was stopped with 1% SDS, and absorbance was measured at 405 nm. For IgG subclass detection, mAbs to human IgG1, IgG2, IgG3 and IgG4 were used at 1:1000 dilution, followed by HRP conjugated goat anti-mouse IgG used at 1:2500 dilution. Finally, 3,3′,5,5′-Tetramethylbenzidine (TMB, Life Technologies) substrate was used for colour development, and the reaction was stopped with 1 M $H_2SO_4$. The absorbance was measured at 450 nm. A pool of Kenyan adult serum with high IgG reactivity to CSP was used as a positive control.

**Mapping antibody epitopes of the NT region.** An array containing 70 biotinylated 15-mer peptides representing the linear sequence of CSP (3D7/NF54 strain) was synthesised (Mimotopes, Australia). Each peptide had a short SGSG spacer between the CSP peptide sequence and the biotin tag (which was added to aid immobilisation), and a 12 aa overlap between peptides. Antibody assays were conducted using previously established approaches[65,66]; 50 μl per well of peptides were added to ELISA plates that were pre-coated with 1 μg/ml streptavidin (Sigma-Aldrich). Saturation of peptide binding was achieved and confirmed by quantification of residual biotin-binding sites on streptavidin using biotin-HRP. followed by blocking with 10% milk, and washing with PBS with Tween 0.05%. Rabbit polyclonal IgG raised against the NT region of CSP was added to each well at a concentration of 1:500. The level of rabbit IgG binding was detected by HRP-conjugated goat anti-rabbit IgG (Millipore, 1:2500) and quantified using the TMB substrate.

**Fcγ receptor-binding assay.** The Fcγ receptor-binding assay was adapted from previously published methods with minor modifications[39]. Briefly, 50 μl of recombinant CSP at a concentration of 1 μg/ml was coated on Maxisorp™ plates and incubated overnight at 4 °C, followed by three washes with PBS-Tween. The plates were then blocked with 200 μl of 1% BSA-PBS at 37 °C for 2 h followed by three washes with PBS-Tween. Serum samples were diluted at 1:50 in PBS-BSA and 50 μl of each sample was added to the plates in duplicate and incubated at room temperature for 2 h, followed by three washes in PBS-Tween. Fifty microliters of biotin-conjugated recombinant soluble FcγRIIa H131 ectodomain dimers (0.2 μg/ml) or FcγRIII V158 ectodomain dimers (0.1 μg/ml) were added to each well and incubated at 37 °C for 1 h followed by three washes with PBS-Tween. For some assays, the R131 allele of FcγRIIa and the F158 allele of FcγRIII was used at the same concentration. This was followed by a secondary HRP-conjugated streptavidin (Thermo Fisher Scientific, 1:10,000) in PBS-BSA at 37 °C for 1 h followed by three washes with PBS-Tween. Finally, 50 μl of TMB liquid substrate was added for 20 min to measure enzymatic reactivity. The reaction was stopped with 1 M $H_2SO_4$ solution. The level of binding was measured as optical density at 450 nm. Pooled human IgG from malaria-exposed Kenyan adults (1:100) and rabbit polyclonal IgG to CSP (1:500) was used as positive controls. Individual sera from naive Melbourne adults (1:50) were used as negative controls.

For FcγR binding to merozoites, 50 μl of merozoites at a concentration of $1 \times 10^7$/ml were coated to Maxisorp™ plates. The FcγR-binding assay was performed as described above, except that PBS without 0.05% Tween were used for washing and purified human IgG from malaria exposed Kenyan adults[32] at a concentration of 1:250 were used for positive controls.

**Antibody-dependent cell-mediated cytotoxicity assay.** ADCC-inducing activity of a selection of serum samples was assessed using the ADCC Reporter Bioassay kit (Promega, Cat. No. G7010) as per the manufacturer's instructions, with the following modifications. CSP-coated beads were opsonized with human serum at 1:50 dilution. Reporter cells were plated at 72,000 cells per well together with CSP-coated fluorescent beads at a 10:1 ratio. After 6 h of incubation, the chemiluminescence substrate was added and luminescence detected and quantified approximately every 5 min for 40 min (CLARIOstar, BMG LabTech). Data presented were recorded about 30 min after the addition of the chemiluminescence substrate.

The level of ADCC was also confirmed in primary NK cells using a previously published method[38]. Briefly, PBMCs containing NK cells were cultured in RPMI1640 supplemented with 10% FCS and 100 IU/ml of interleukin-2 (IL-2) overnight. CSP-coated beads were opsonized with human serum at 1:50 dilution, then co-cultured with the primed PBMCs in the present of anti-CD107a-AF647 (H4A3, BD Bioscience) for 1 h. Subsequently, brefeldin A (Sigma-Aldrich) and protein transport inhibitor (BD Bioscience) were added to the co-culture at a concentration of 5 μg/ml and 1/1500, respectively, and continued to incubate for 3 h. After incubation, the cells were centrifuged at $300 \times g$ for 4 min at 4 °C and stained with anti-CD3-APC-H7 (SK7, BD Bioscience, 1:33), anti-CD56-PE-Cy7 (B159, BD Bioscience, 1:33) and anti-CD16-BV421 (3G8, BD Bioscience, 1:20). NK cells were defined as CD3−, CD56+ lymphocytes and the level of ADCC was quantified as the percentage of NK cells with CD107a staining by flow cytometry (LSR Fortessa X-20, BD Bioscience) (Supplementary Fig. 9).

**Data analysis.** Statistical analyses were performed using Prism version 7 (GraphPad Software Inc) and STATA version 13.1 (STATACorp). Statistics test and $P$ values are indicated in figure legends. Correlations between phagocytosis, FcγR binding and IgG levels were evaluated using Spearman's correlation coefficients and multiple linear regression models. The differences in levels of phagocytosis, FcγR binding and IgG levels across age groups and different opsonization conditions were assessed using the one-way ANOVA with multiple comparisons. To evaluate the antibody functional activities irrespective of total IgG titre to CSP, FcγR binding and phagocytosis efficiencies were calculated as the level of FcγR binding or the level of opsonic phagocytosis relative to IgG reactivity to CSP. Spearman's correlation and Mann–Whitney test were used to analyse antibody functional efficiencies among children and adults. Two-way ANOVA was used to compare the opsonic phagocytosis by neutrophils and monocytes in the whole-leukocyte assays. Kruskal–Wallis test and Mann–Whitney test were performed for data that were not normally distributed. All statistical analyses were performed as two-sided tests. Samples were classified as positive for antibody if reactivity was greater than the mean + 3 standard deviations of the values of the malaria non-exposed controls.

**Reporting summary**. Further information on research design is available in the Nature Research Reporting Summary linked to this article.

## Data availability

Data used to generate figures are available from the authors on request. Databases that were analysed in the preparation of this paper and presented in figures and tables are available from the corresponding author upon reasonable request and pending agreement from relevant ethics committees (for clinical data). *P. falciparum* sequences were obtained from the public database at PlasmoDB.org.

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

## Acknowledgements

We thank the study participants and staff of the Kenyan Medical Research Institute for samples from Chulaimbo cohort and Kanyawegi cohorts. *P. falciparum* sporozoites, and mAb 2H8 were kindly provided by PATH Malaria Vaccine Initiative (MVI). The mAb 3D11 and hybridoma were kindly provided by Miguel Prudêncio of the Instituto de Medicina Molecular. *P. berghei* parasites expressing *P. falciparum* CSP were provided by Chris Janse, Shahid Khan, and Hans Kroeze of Leiden University Medical Center. Covalent coupling of antigen to fluorescent beads was greatly supported by Paul Ramsland of RMIT, Paul Sanders of Burnet Institute and Geoffrey Pietersz of the Baker Heart and Diabetes Institute. We also thank Michelle Wong of Burnet Institute for being our phlebotomists and Emily Locke for helpful feedback and advice. This study was also supported by AMREP Flow cytometry core facility. This work was supported by the National Health and Medical Research Council (NHMRC) of Australia (Program Grant and Senior Research Fellowship to JG Beeson; Project Grants to J.G.B., GI McFadden, and PM Hogarth; Career Development Fellowship to M.J.B.), PATH-MVI (grant to J.G. B.), CASS Foundation (grant to G.F.), Australian Research Council Laureate Fellowship to GI McFadden, and NIH International Centers for Excellence in Malaria Research (U19AI129326). The Burnet Institute is supported by the NHMRC for Independent Research Institutes Infrastructure Support Scheme and the Victorian State Government Operational Infrastructure Support. J.G.B., G.F., M.J.B., J.A.C., SC and L.K. are members of the NHMRC Australian Centre for Research Excellence in Malaria Elimination.

## Author contributions

J.G.B., G.F. and B.D.W. led the study design with input from P.M.H., G.I.M., L.K., J.A.C. and other authors; G.F., B.D.W., L.K., P.B., D.M.L., S.C., V.M. and A.C. conducted experiments and interpreted the results; G.F., P.B. and J.A.C. conducted data analysis with input from J.G.B., L.K. and M.J.B.; J.G.B., B.D.W., P.M.H., J.W.K., A.E.D., K.C., G.I.M., A.C., V.M., D.R.D. and R.J.C. provided or generated key reagents, data and/or resources; G.F., J.G.B., L.K., J.A.C. and D.R.D. contributed to paper preparation with input from all other co-authors.

## Competing interests

The authors declare no competing interests.
