## [Peer Review File · Nature Communications]

Reviewers' comments:

Reviewer #1 (Remarks to the Author):

Feng et al. describes an interesting observation related to the presence of Fc-effector functional antibodies in malaria exposed individuals. Focused specifically on the importance of the circumsporozoite protein, the major antigen in advanced vaccine programs, the authors adapt a bead-based phagocytic assay that has been used for vaccine evaluation for the analysis of CSP-specific antibody functions in malaria exposed populations. In parallel the authors adapt a frozen-sporozoite phagocytic assay to look at activity against the actual parasite. The authors demonstrate the presence of phagocytic antibodies to this antigen in malaria exposed, but not unexposed individuals. Moreover, the authors suggest that the dominant function of these antibodies in whole blood occurs via neutrophils, rather than monocytes. This activation occurs via both FcγR2 and FcγR3, but blocking of FcγR3 results in a more dramatic reduction in antibody mediated phagocytosis in neutrophils. Moreover, the authors map the response across CSP, pointing to higher levels of phagocytic activity to the repeat region compared to other regions in the antigen. Using cross-sectional cohorts, the authors then show that this activity is lower in children compared to adults, suggesting that this activity may evolve with exposure, potentially contributing to natural immunity over time. Thus, collectively, using complementary approaches, an array of reagents (patient samples and immunized animal serum), and unique cohorts, the authors conclude that neutrophil-recruiting antibodies may represent an important Fc-effector function in malaria immunity.

While the concept is very exciting for vaccine development, and may add tremendous value to understanding protective immunity against malaria, several questions arise that create uncertainty around the conclusions.

1. One of the central conclusions of the manuscript is that "opsonic phagocytosis of sporozoites is predominantly mediated by neutrophils in peripheral blood". This conclusion is largely made based on experiments showing that using a whole leukocyte phagocytosis assay that "the number of beads phagocytosed by neutrophils was much higher than that by monocytes". However, given that relatively low number of monocytes present in the peripheral blood compared to neutrophils, it is likely that the observed low monocyte phagocytosis is due to the low number of monocytes present in the assay and that the excess of neutrophils is outcompeting any potential monocyte phagocytosis. Additionally, direct comparison is also difficult due to the fact that purification of monocytes and neutrophils is done quite differently. Furthermore, for all phagocytosis assays (using cell lines or primary cells against any antigen), phagocytosis was allowed to proceed for 15-20 minutes. While phagocytosis by neutrophils is extremely fast, monocyte-mediated phagocytosis is orders of magnitude slower. Combined with the increased number of neutrophils likely present in the assay, it is not surprising that more phagocytosis mediated by neutrophils is observed and that conclusions may be skewed by kinetics and cell frequencies. Some efforts to account for these issues would help with the conclusions.

2. Whole blood is derived from healthy non-endemic donors, which may have significantly different FcR expression on neutrophils and monocytes. The overall FcR expression on the donors is not illustrated, rendering it difficult to interpret the results. Given that the authors report that the inhibitory effect of phagocytosis by blocking FcγR2a was highly variable across experiments, how much variation is there from donor to donor? Moreover, because 3a is expressed at low levels, and signaling is not shown for 3a, is it possible that binding to 3b may simply facilitate clustering of FcγR2s, resulting in neutrophil activation. Dose down curves would provide the data required (reciprocal dilutions for fcγ2 and fcγ3 blocking antibodies) to help substantiate the conclusions.

3. The THP1 and primary monocyte difference in the presence or absence of serum is interesting, but there are several concerns that may impact the interpretation of the results- and should be clarified in the text. First, FcγR1 is tonically occupied by antibodies, even those found in serum containing media. It is difficult to therefore understand the results shown here. Importantly, while only a few malaria-related THP1-based studies have been carried out, THP1-based phagocytosis

assays are widely used in other diseases. In these diseases, traditional THP1-based phagocytosis assays are nearly always carried out in the presence of serum, so it is unclear why the authors chose to run their assays in serum-free medium. Moreover- this difference must be clarified in the results section- to clarify why the results here are unlikely to ever reflect what happens naturally with monocytes in vivo or ex vivo (where they are likely to still be occupied). Finally, THP1 cells vary dramatically by passage number. It is essential to control for this parameter and show the results over up to 30 different passages to ensure the conclusions are robust. The authors should omit broad conclusions if precision is not applied to describe the effects of media, FBS lots, and passage number given the importance of all of these factors in defining the functional activity of the THP phagocytic assay.

4. While the analysis of the malaria-exposed cohorts is intriguing, the study is largely observational and correlative, and very few significant conclusions can be drawn. Furthermore, the clinical description of these samples is limited, which complicates many of the analyses. Are these individuals all asymptomatic? Do the individuals have equivalent parasitemia if infected? Do these antibody functions precede infection and correlate with enhanced protection/control? The overall importance of these functional antibodies are a little confusing given the lack of information on the cohort malaria status. This may be most relevant among the children where substantial variation is observed in functional antibody activity. Additional analysis related to parasitemia or disease would be quite important.

5. Finally, the relevance of CSP as an antigen is a little confusing. While CSP is an important vaccine antigen, its role in natural immunity is quite unclear. Natural immunity is generally associated with the evolution of broad humoral immune responses, but have been classically associated with blood-stage immunity. Thus, it would be quite interesting to determine whether similar responses observed to the blood stage antigens?

Minor concerns:

1. It is unclear why a number of experiments to full length CSP were carried out using polyclonal rabbit serum rather than readily available human monoclonal or polyclonal serum (e.g., Figure 1 D, E).
2. There is a general lack of consistency in the methodology used in Figure 1 that makes comparisons between the panels impossible. Both the target antigen as well as the assay conditions (e.g., serum or no serum) vary from panel to panel.
3. Given the binding of antibodies to Fcgr3, the inclusion of ADCC activity seems like a natural step. However, the selected method is (using a Fcgr3a-expressing reporter cell line) not a direct ADCC assay and just reports Fcgr3 signaling. This should be clearly specified and not referred to as proof of cytotoxicity.
4. Neutrophils mediate an array of functions. Beyond phagocytosis, do the functional antibodies trigger additional neutrophil functions (cytokines?)? Do these change with antibody function, FcR usage, IgG titer dependence? Do these downstream inflammatory responses track with protection or inflammation/fever? Enhanced resolution with cohort characteristics may provide enhanced resolution on the immune protective functions of these functional antibodies.

Reviewer #2 (Remarks to the Author):

NCOMMS-19-29433

- Key results: Please summarise what you consider to be the outstanding features of the work.

This work by Feng et al aims to characterise the role of Fc γ R and opsonic phagocytosis in anti-sporozoite-mediated immunity to malaria. Using a range of novel assays to probe samples from a region of medium to high transmission, the role of and interactions between different phagocyte populations and Fc γ Rs is explored. The observation that most opsonisation of sporozoites is

neutrophil mediated is novel as is the definition of which Fc γ Rs are utilised. These data are a welcome addition to the field as mechanisms of sporozoite-mediated immunity are poorly understood compared with immunity to blood-stage for example. Although the RTS,S vaccine is the first vaccine to show significant efficacy in the field and targets the sporozoite stage, until recently little immunology has been undertaken to understand the mechanisms of immunity associated with vaccine efficacy.

- **Validity:** Does the manuscript have flaws which should prohibit its publication? If so, please provide details.

No, overall, the approach is logical, and the experimental methodology is sound.

- **Originality and significance:** If the conclusions are not original, please provide relevant references. On a more subjective note, do you feel that the results presented are of immediate interest to many people in your own discipline, and/or to people from several disciplines?

The conclusions are original to the best of my knowledge and build on a growing body of work from this and other groups aiming to define protective mechanisms of anti-CSP antibodies. This is a very important area for development of malaria vaccines and is really gathering pace. This study could potentially make an important contribution to that.

- **Data & methodology:** Please comment on the validity of the approach, quality of the data and quality of presentation. Please note that we expect our reviewers to review all data, including any extended data and supplementary information. Is the reporting of data and methodology sufficiently detailed and transparent to enable reproducing the results?

This section is where the major weakness of this work lies. Although the methods used are mostly appropriate, the presentation of the data is not and could be improved substantially. I found many of the figures quite difficult to follow and it wasn't always clear what was being shown. In particular, much of data is presented as bar charts, without the number of replicates indicated and so it is impossible to determine the precision of any of the data or how variable the results between replicates were. For the flow-based assay, no information is provided about the total number of events collected. For the OPA, it would be useful to know how many replicate wells were tested or whether only a single well was used. In the legends of the figures 2E and 3A, the authors state "selected individuals are shown as examples" without explaining how the examples were chosen, which could be interpreted as showing only the most favourable data points. In Figure 5, no information is given as to how many participants are included in each age-group. Throughout the paper, p values are presented without defining the test used (although this is stated in the methods section). There was too little information given both on the figures and in the legends to interpret the data without referring back to the text. Some of the figures could be combined together e.g. 4A and B.

- **Appropriate use of statistics and treatment of uncertainties:** All error bars should be defined in the corresponding figure legends; please comment if that's not the case. Please include in your report a specific comment on the appropriateness of any statistical tests, and the accuracy of the description of any error bars and probability values.

The statistical tests described are appropriate, however there are numerous references to data in the text that are not supported by a formal analysis e.g. line 378- "Neutrophils had substantially lower opsonic phagocytosis of both CSP-beads and *P. falciparum* sporozoites in the presence of either Fc γ RIIa or Fc γ RIII blockers (Figure 2A and B)". No indication of whether a statistical test was applied or not.

- **Conclusions:** Do you find that the conclusions and data interpretation are robust, valid and reliable?

This is very hard to assess because the presentation of the data is not clear enough.

- **Suggested improvements:** Please list additional experiments or data that could help

strengthening the work in a revision.

- References: Does this manuscript reference previous literature appropriately? If not, what references should be included or excluded?

The manuscript is well referenced.

- Clarity and context: Is the abstract clear, accessible? Are abstract, introduction and conclusions appropriate?

The abstract and introduction are clear and well-written.

- Please indicate any particular part of the manuscript, data, or analyses that you feel is outside the scope of your expertise, or that you were unable to assess fully.

Specific comments to authors:

1. What was the viability of the cryopreserved *P. falciparum* sporozoites? My understanding is that most are dead, so 50,000 Pf sporozoites would be very different to 50,000 freshly dissected *P. berghei* sporozoites. Why were different dyes used to label the two species of sporozoites? Does the cytometer have the same sensitivity for the two dyes, such that the signal is comparable?
2. In Figure 1A, the number of replicate wells tested should be stated and the statistical test used stated. In B and C the dotted lines should be described in the legend. In C, relatively few data points are shown to conclude that the correlation between beads and Pf sporozoites was acceptable. The legend mentions high, intermediate and low responses, but only two populations are discernible.
3. In Figure 1 D and E, these data would be better represented with line graphs and suitable error bars, I would suggest that for only 6 replicates, a non-parametric average and error would be more suitable.
4. SF2- the lower panels need to be more clearly labelled as it was hard to determine what they related to (neutrophils on the left and monocytes on the right I think).
5. Fig2A and B, number of replicates is not given and spread of data cannot be assessed, so a different type of graph would be more appropriate. These two panels could be combined. 2E How were the individuals shown selected? 2F could the authors comment on the strength of the correlation with a r of 0.4. It looks like a relatively weak association.
6. Figure 3. Again, how were examples selected for 3A? The authors state "Overall 67.3% and 61.5% of samples ($n=104$) were considered positive for $Fc\epsilon\geq RIIa$ and $Fc\epsilon\geq RIII$ binding". How was positivity determined?
7. I have no sense of how reproducible the assay is and how many cells were acquired in order to determine the phagocytosis index? If each data point is just a single well of 5000 cells, how many of those were actually run on the cytometer? Was extracellular fluorescence from sporozoites that were not phagocytosed, but stuck to the outside of cells quenched?
8. Figure 5, number of individuals per age-group is not indicated. Statistical tests used is not described.
9. Figure 6. It is unclear how corrections for multiple analyses was performed, although the authors allude to a correction in the methods section. Given that the multivariate analysis is a much more suitable way to interpret all of these associations, could this figure be moved to the supplementary and Table S4 presented in the main text.

Katie Ewer

Reviewer #3 (Remarks to the Author):

While we have known for decades that CSP-specific antibodies can protect people and mice, at least in part through their impact on sporozoite motility, it remains an open question whether

these antibodies also have indirect sporozoite neutralization activity via the Fc portion of antibody. In this study the authors use in vitro phagocytosis assays (with neutrophils or the monocyte cell line THP1), combined with different sources of antisera (rabbit, mouse, or Kenyan adults), and different sources of the target antigen CSP (sporozoites, peptides and recombinant protein on beads) to determine that antisera specific for CSP can enhance the phagocytic uptake of CSP coated beads and antibody-opsonized sporozoites by neutrophils. This is not surprising as most targets in the presence of antibodies would be taken up in greater amounts by neutrophils in a small well where the optimal pathogen:neutrophil ratio is used and the neutrophils are given sufficient time to recognize and take up the target. What is never addressed is whether this has any relevance to malaria infection. Figures 3, 5, and 6 show that sera from people in a malaria-endemic area are capable of binding to CSP-coated beads and Fc receptors, particularly FcRIII, and be taken up by neutrophils in vitro. The relevance of CSP-coated beads to sporozoites is weak. Sporozoites are one of the fastest eukaryotic cells, capable of easily traversing through phagocytic cells, they are not an inert bead. There are, however, a few experiments with sporozoites, both cryopreserved *P. falciparum* and *P. berghei* expressing PfCSP in Figures 1F&G, 2B, 4A,B&D. None of the experiments with sporozoites have statistics and it looks like the background is between 5-7% phagocytosis (i.e. in the presence of an irrelevant antibody) while the signal is 10-15%. Since PbPfCSP are rodent malaria parasites, the tools exist to test whether any of their findings have any relevance in vivo. Without in vivo experiments, the relevance of the in vitro activity of antibody, particularly when using CSP-coated beads, is not clear. Indeed, the significance of these in vitro findings is questionable given the previous demonstration that adults in malaria endemic areas have no functional pre-erythrocytic stage immunity (TM Tran et al Clin Infect Dis 2013).

Specific Comments:

The most interesting and relevant experiments in this paper have no statistics and the data, with the exception of Fig 4A, do not look significant. Statistics would be needed for Figures 1F, 1G, 2B, 4A, 4B, 4D.

Controls vary from one experiment to another. Most frequently its unopsonized beads or parasites. This is an important control but more relevant would be adding non-specific sera or antibody. When this was done in Figure 4B (MAB 3D11), the true background looks like its 5 to 7%.

The concentration of mAb 2H8 used for the opsonization experiment in Fig 4B is not stated in the paper. Do the parasites still move? Are the neutrophils merely mopping up dead immobilized parasites?

Experiments in which different CSP peptides are immobilized onto beads and then used in phagocytosis and FcR binding assays could be skewed by the amount of peptide that binds to the beads. Frequently this can be sequence dependent. Thus, it is essential to measure this and attempt to have equivalent amounts of peptide per bead for each peptide.

The authors demonstrate that the monocyte cell line THP1 is not good for these studies since opsonization/ phagocytosis is inhibited by human serum. The experiments with THP1 should therefore all be moved to the supplemental as beginning the paper with a figure where most of the data is with a cell line that is later shown to be problematic starts the paper off on a weak footing.

Unfortunately there are issues with cryopreserved Pf sporozoites. Once sporozoites are purified, frozen and then thawed, the majority are dead. This is made clear in the clinical trials with these sporozoites, which have necessitated much higher doses to induce immunity than what was originally thought based on irradiated sporozoites delivered by mosquito bite. Freshly dissected PbPfCSP sporozoites are more robust if one is interested in looking at phagocytosis of live sporozoites. For this reason, it would be important to perform the FcR blocking experiment (Fig 2B) and the opsonization/phagocytosis experiments with polyclonal and monoclonal antibodies (Fig 4A&B) with PbPfCSP parasites.

References:

Ref 5 does not show that sporozoites are inoculated into a pool of blood. There is no reference I

am aware of that shows this. However, the following references show that sporozoites are inoculated into the skin – Matsuoka et al. *Parasitol Int* 2002; Sidjanski et al 1997 *AmJTropMedHyg*.

Refs 11-16 which show that antibodies to CSP inhibit motility should include two recent in vivo studies: Flores-Garcia et al *mBio* 2018; Aliprandini et al *Nat Microbiol* 2018.

Its stated that little is known about functional regions of the CSP. Actually a functional region in the N-terminus was shown in Coppi et al *JEM* 2005 and 2011, and should be mentioned, especially given their data with the N-terminal peptide.

Response to Reviewers

General comments:

We thank the reviewers for their time taken to review our manuscript and for providing many helpful suggestions and identifying issues that needed clarifying or strengthening. We have responded to all points raised by the reviewers below. We have undertaken a major revision of our manuscript, including the addition of new experimental data and analysis as well as re-writing and editing text throughout the introduction, results and discussion sections. As identified by the reviewers, some of the key points and findings were not sufficiently clear in the previous manuscript and we hope that the key findings are presented more clearly now and the overall flow of the manuscript is improved.

We have included important new experimental data to strengthen and extend our findings:

1. We have completed repeat assays and included further replicates in our analysis. We have now indicated statistical significance indicated for all major observations reported in the text, figures, and tables, and statistical tests used for each p value cited are indicated in the figure legends.
2. One of the significant findings from our work was that the N-terminal region of CSP was a target of functional antibodies, which is significant since the N-terminal region of CSP is not included in the RTS,S vaccine, or in the other leading CSP vaccine known as R21 (currently in clinical trials). Since we found that antibodies to the N-terminal region had strong functional activity, we have now successfully mapped the epitopes of these antibodies to further enable novel vaccine design aimed at achieving greater vaccine efficacy (Figure 4). This has not been reported previously.
3. We have included data using our recently established assays using fresh primary NK cell. Our findings show, for the first time, that antibodies to CSP can activate NK cells, and activation correlates with binding to FcγRIII. This has not been previously demonstrated for immunity to sporozoites. (Figure 2E, 2F, 2G and 2H)
4. We have expanded our analysis of monocyte phagocytosis to identify the classical monocyte subset as the most active subset for phagocytosis. Again, this has not been previously reported for immunity to sporozoites. (Figure 1D)
5. We have included new data on the acquisition of antibodies that promote phagocytosis of merozoites by neutrophils, for comparison with acquisition of functional antibodies to sporozoites (Figure 5B, 5E and 5F). These data show that the acquisition of functional antibodies to sporozoites among young children is poor in comparison to the acquisition of functional antibodies to merozoites. In contrast, adults have substantial functional antibodies to sporozoites and merozoites.
6. Additional data have been included from phagocytosis assays where FcγR blocking antibodies have been titrated from high to low concentration to further support conclusions about the roles of specific FcγRs (Figure S6).
7. To understand the significance of these functional antibodies in vivo and in vaccine-mediated immunity, we applied our new assays to a phase 1/2a trial of RTS,S. This showed that antibodies that promoted neutrophil phagocytosis and binding of FcγRIII and FcγRIIa were significantly higher among protected than non-protected subjects in the vaccine trial. We have now posted a complete draft of a manuscript reporting these findings on the pre-print server BioRxiv (<https://doi.org/10.1101/851725>; with 6 figures and Supplementary Material). We hope these findings provide the reviewers

with confidence that our new insights into FcγR mediated mechanisms of immunity have substantial relevance to vaccine development and evaluation.

Reviewer #1 (Remarks to the Author):

Feng et al. describes an interesting observation related to the presence of Fc-effector functional antibodies in malaria exposed individuals. Focused specifically on the importance of the circumsporozoite protein, the major antigen in advanced vaccine programs, the authors adapt a bead-based phagocytic assay that has been used for vaccine evaluation for the analysis of CSP-specific antibody functions in malaria exposed populations. In parallel the authors adapt a frozen-sporozoite phagocytic assay to look at activity against the actual parasite. The authors demonstrate the presence of phagocytic antibodies to this antigen in malaria exposed, but not unexposed individuals. Moreover, the authors suggest that the dominant function of these antibodies in whole blood occurs via neutrophils, rather than monocytes. This activation occurs via both FcγR2 and FcγR3, but blocking of FcγR3 results in a more dramatic reduction in antibody mediated phagocytosis in neutrophils. Moreover, the authors map the response across CSP, pointing to higher levels of phagocytic activity to the repeat region compared to other regions in the antigen. Using cross-sectional cohorts, the authors then show that this activity is lower in children compared to adults, suggesting that this activity may evolve with exposure, potentially contributing to natural immunity over time. Thus, collectively, using complementary approaches, an array of reagents (patient samples and immunized animal serum), and unique cohorts, the authors conclude that neutrophil-recruiting antibodies may represent an important Fc-effector function in malaria immunity.

While the concept is very exciting for vaccine development, and may add tremendous value to understanding protective immunity against malaria, several questions arise that create uncertainty around the conclusions.

We thank the reviewer for their encouraging comments and we have responded to their comments and points raised in order to address areas of uncertainty and clarify our findings. We have now added substantial new data, conducted additional analysis and repeat assays to demonstrate reproducibility and statistical significance of all findings, and made major revisions to the manuscript to address the issues raised.

1. One of the central conclusions of the manuscript is that “opsonic phagocytosis of sporozoites is predominantly mediated by neutrophils in peripheral blood”. This conclusion is largely made based on experiments showing that using a whole leukocyte phagocytosis assay that “the number of beads phagocytosed by neutrophils was much higher than that by monocytes”. However, given that relatively low number of monocytes present in the peripheral blood compared to neutrophils, it is likely that the observed low monocyte phagocytosis is due to the low number of monocytes present in the assay and that the excess of neutrophils is outcompeting any potential monocyte phagocytosis. Additionally, direct comparison is also difficult due to the fact that purification of monocytes and neutrophils is done quite differently. Furthermore, for all phagocytosis assays (using cell

lines or primary cells against any antigen), phagocytosis was allowed to proceed for 15-20 minutes. While phagocytosis by neutrophils is extremely fast, monocyte-mediated phagocytosis is orders of magnitude slower. Combined with the increased number of neutrophils likely present in the assay, it is not surprising that more phagocytosis mediated by neutrophils is observed and that conclusions may be skewed by kinetics and cell frequencies. Some efforts to account for these issues would help with the conclusions.

Response: A number of points were raised by the reviewer, which we will address in turn.

i) Phagocytosis is predominantly by neutrophils: To understand the potential roles of different cell types for phagocytosis of sporozoites, we conducted assays using whole blood as this would best represent conditions in vivo where all cell types are present. We quantified the phagocytosis activity of neutrophils and monocytes using CSP-coated beads and live intact sporozoites. This demonstrated that neutrophils were much more active phagocytes than monocytes. We clarify that we quantified the phagocytosis rate (number of beads or sporozoites that were phagocytosed per cell) by neutrophils and monocytes. Therefore, the higher activity of neutrophils is not simply explained by their higher abundance in blood compared to monocytes. What we found is that a higher proportion of the neutrophil population, compared to the monocyte population, phagocytosed sporozoites or CSP-beads. This can be seen in Fig 1A, 1B, and 1C. We explored different phagocyte:bead ratios and different antibody concentrations and found that the rate of phagocytosis by neutrophils was always higher than monocytes. This is an important finding that has not been reported previously in malaria immunity. In the revised manuscript we have added additional text and descriptions to make this point clearer in the results and discussion sections (Lines 357-362).

ii) Reviewer comment that 'purification of monocytes and neutrophils is done quite differently': We clarify that assays to quantify the relative role and activity of different cell types in phagocytosis of sporozoites and CSP-beads were performed using whole blood in which all potential phagocytes are present. This would best represent conditions in vivo. Purification of individual cell types was only performed to support our findings in whole blood assays. Therefore, the issue of different cell purification methods potentially impacting on activity of monocytes and neutrophils is not a factor here. This point was not sufficiently clear in the previous version of our manuscript and we have made substantial revisions in the text and figure organisation to make this clearer.

iii) Time allowed for phagocytosis: Regarding the time allowed for phagocytosis, there were several considerations providing the rationale for our 15-20 min incubation. First, we had previously evaluated the timing of phagocytosis by monocytes and found that extending beyond 20 mins incubation did not markedly increase phagocytosis rate. In the literature, others also use incubation of times of 20-30 mins for phagocytosis. A second consideration is that efficient phagocytosis most likely needs to occur in this timeframe for it to be a biologically significant mechanism of immunity. Once sporozoites enter the circulation from the dermal inoculation site, phagocytosis will need to be occurring quickly for clearing sporozoites from the circulation, thereby reducing or preventing liver infection.

General comments and new data: Our data suggest that neutrophils are much more active in phagocytosis of sporozoites than monocytes, and their greater abundance would also contribute to their greater significance in immunity. This finding has not been established in the published literature and the role of neutrophils has been greatly understudied in malaria immunity more broadly. We also acknowledge that monocytes are still likely to be making a

significant contribution to clearing sporozoites through phagocytosis, despite their lower phagocytic rate than neutrophils - we have modified the text accordingly. To better understand the role of monocytes we have also now included interesting new data in which we have determined which monocyte subsets are most active in phagocytosis. This shows that the classical subset is by far the most active in phagocytosis (Figure 1D)

2. Whole blood is derived from healthy non-endemic donors, which may have significantly different FcR expression on neutrophils and monocytes. The overall FcR expression on the donors is not illustrated, rendering it difficult to interpret the results. Given that the authors report that the inhibitory effect of phagocytosis by blocking Fcgr2a was highly variable across experiments, how much variation is there from donor to donor? Moreover, because 3a is expressed at low levels, and signaling is not shown for 3a, is it possible that binding to 3b may simply facilitate clustering of FcR2s, resulting in neutrophil activation. Dose down curves would provide the data required (reciprocal dilutions for fcgr2 and fcgr3 blocking antibodies) to help substantiate the conclusions.

Response: The reviewer raised several points, which we have addressed in turn

i) Expression of FcγR types on different cell types. The expression of FcγR types on monocytes, neutrophils, and NK cells has been extensively characterised and reported in the literature and therefore we have not provided a characterisation of FcγR expression in this paper. Instead we conducted an analysis of the functional roles FcγRs play in phagocytosis by using FcγR blocking with live immune cells in phagocytosis assays. These data (see Fig 2A, 2B, and 2C) are important because they indicate which FcγRs are being used, not just which FcγRs are being expressed. Additionally, we examined binding interactions between antibodies to CSP and specific FcγRs using novel FcγR probes (see Fig 2E, E, G, H, Fig 3A and B, and Fig 4E). In an otherwise healthy person, FcγR expression is largely the same across populations. While FcγR expression can vary in different disease states (influenced by a range of activating and inflammatory mediators), evaluating these effects is separate question to the aims of our study and would require additional carefully designed studies of sufficient size and clinical case definition to address.

ii) Variability in FcγRIIa blocking: Our prior manuscript was not worded clearly and we have now addressed that in the revision. We had suggested there was more variability in the effects of FcγRIIa blocking. However, we have conducted multiple repeat experiments and observed only modest inhibitory effect of FcγRIIa blocking with neutrophils (see Fig 2A and B, and Supplementary Fig S6). We have updated the figures and included additional data in the Supplementary file (Fig S6). Figures show error bars indicating variance and the number of experiments is stated in the figure legends, and p values with statistical test are now clearly stated.

iii) Role of FcγRIII and need for 'dose down curves': We have now provided additional data on titration of FcγR blocking antibodies (tested alone and in combination) as suggested (see Supplementary Figure S6). These additional data, and the data provided in Figs 2A-C, show that the greatest inhibition of phagocytosis is achieved with the FcγRIII blocking antibody, and there is only partial inhibition by blocking FcγRIIa alone. When blockade of FcγRIII and FcγRIIa is combined, there is greater inhibition of phagocytosis supporting a role for FcγRIII and FcγRIIa. If FcγRIIIb was mainly acting by facilitating clustering of FcγRIIa, then we would expect to see much greater inhibition of phagocytosis by the FcγRIIa blocking

antibody. Overall, our data support an important role for FcγRIII and an additional role for FcγRIIIa. We have now included statistical testing and p values for all comparisons and test conditions to support these conclusions. We have revised the wording to make the description and discussion of findings clearer and consider the reviewers point.

3. The THP1 and primary monocyte difference in the presence or absence of serum is interesting, but there are several concerns that may impact the interpretation of the results- and should be clarified in the text. First, FcγR1 is tonically occupied by antibodies, even those found in serum containing media. It is difficult to therefore understand the results shown here. Importantly, while only a few malaria-related THP1-based studies have been carried out, THP1-based phagocytosis assays are widely used in other diseases. In these diseases, traditional THP1-based phagocytosis assays are nearly always carried out in the presence of serum, so it is unclear why the authors chose to run their assays in serum-free medium. Moreover- this difference must be clarified in the results section- to clarify why the results here are unlikely to ever reflect what happens naturally with monocytes in vivo or ex vivo (where they are likely to still be occupied). Finally, THP1 cells vary dramatically by passage number. It is essential to control for this parameter and show the results over up to 30 different passages to ensure the conclusions are robust. The authors should omit broad conclusions if precision is not applied to describe the effects of media, FBS lots, and passage number given the importance of all of these factors in defining the functional activity of the THP phagocytic assay.

Response: We apologise that this section on THP-1 phagocytosis activity and the influence of serum was not clear in our previous manuscript. We have now substantially revised the text and clarified the conditions to which we were referring. The reviewer raised several points, which we have answered in turn:

i) Description of serum-free conditions, and comparisons of conditions: When we described 'serum-free', we meant free of human serum. This was not made sufficiently clear in the previous manuscript. The assay medium still contained fetal calf serum (FCS), which is standard practice in the published literature for THP-1 culture and performing phagocytosis assays. However, bovine IgGs do not interact with human FcγRs, and therefore would not interact with human FcγRI as human monomeric human IgG does. In our experiments we demonstrate that the phagocytosis rate of THP-1 cells is much lower when human serum (containing human IgG) is present in the assays compared to standard conditions using FCS alone (containing bovine IgG). We have now revised the manuscript and made this explicitly clear in the text and figure legends.

ii) Clarification of results and discussion. In response to this point, we have made substantial revisions to the manuscript to ensure the results presented are clear and to better consider the implications of these findings. Further, we have modified and clarified the discussion of these findings in the manuscript. Because the difference in activity of THP-1 cells in the presence/absence of human serum is not a major point, we have reduced and simplified the text and discussion, and moved some of the THP-1 results figures to the Supplementary file (Figure S4). We believe the inclusion of some data using THP-1 cells is important because the THP-1 cell model is widely used and our data suggest that the model has limitations in representing FcγR-mediated mechanisms. Assays with THP-1 cells are typically conducted in the presence of FCS alone rather than FCS together with human serum, which could be giving misleading results.

iii) *Regarding passaging of THP-1 cells:* We use THP-1 cells for up to 10 passages, and we have previously shown that activity does not substantially vary for up to 10 passages.

4. While the analysis of the malaria-exposed cohorts is intriguing, the study is largely observational and correlative, and very few significant conclusions can be drawn. Furthermore, the clinical description of these samples is limited, which complicates many of the analyses. Are these individuals all asymptomatic? Do the individuals have equivalent parasitemia if infected? Do these antibody functions precede infection and correlate with enhanced protection/control? The overall importance of these functional antibodies are a little confusing given the lack of information on the cohort malaria status. This may be most relevant among the children where substantial variation is observed in functional antibody activity. Additional analysis related to parasitemia or disease would be quite important.

Response: There are several related points raised, which we will answer in turn:

i) Purpose of including clinical samples: The major focus of our manuscript is on the mechanisms and targets of FcγR-mediated immunity against sporozoites rather than evaluating associations with protection. This includes understanding specific cell types, FcγR types, antibody targets, and antibody properties. Our manuscript has been revised to make the focus clearer. The importance of antibodies for immunity to sporozoites has been established through multiple studies in humans and experimental animals over several years. This includes demonstrations that antibodies to the central repeat region of CSP can protect in animal models from malaria infection. Prior studies in Kenya, our study population, have reported associations between reduced risk of infection and antibodies to CSP and sporozoite antigens (John CC et al, AJTMH 2005; Offeddu V et al Frontiers Immunol 2017). However, despite this knowledge of the protective functions of antibodies, the functional mechanisms of these antibodies remain poorly understood and require new insights, which we address in our manuscript. The analysis of responses in malaria-exposed Kenyan adults and children was primarily designed to study FcγR-mediated functional activity of human antibodies and determine the acquisition of FcγR-mediated functional antibodies and the prevalence of these functional antibodies among adults who have had substantial exposure to malaria in a high transmission area. To understand the role of FcγR-mediated functions, we needed to study human antibodies. This cannot be easily addressed in mouse models because of fundamental differences in FcγR types expressed and IgG subclasses and their properties between mice and humans (described further below). Therefore, the use of samples from human clinical studies was crucial to addressing the aim of understanding antibody functional mechanisms.

ii) The clinical description of the cohort samples was limited: In the revised manuscript we have now included more information on the Kenyan clinical cohorts and the population setting in the methods section (see Lines 111-119) and we have also included details in Supplementary Table S1. This outlines that the subjects were asymptomatic when enrolled and a proportion had active parasitemia, which is typical in a malaria-endemic setting.

iii) Is there a relationship between antibodies and parasitemia in the cohort; need for additional analysis? As noted above, the major focus of our manuscript is on the mechanisms and targets of FcγR-mediated immunity and we have revised the manuscript to give it a clearer focus, including how it addresses current knowledge gaps. The clinical study was not designed to investigate associations between antibodies to sporozoites and

protection against infection. However, we did explore potential protective associations in the adult cohort as the subjects were screened for parasitemia one month after sample collection. For this analysis, we included only individuals with negative blood smear at baseline sample collection and not subjects who were already infected. Within this group, higher level of FcγRIII binding efficiency by antibodies was associated with a reduced risk of being infected one month later, including age and sex as potential confounders (OR=0.469, 95% CI 0.267~0.825, P=0.009; logistic regression). However, we are reluctant to include these analyses in the paper as the study was not designed to assess protective associations.

Instead, the best evidence to support a role for these antibody functions in human immunity would come from an analysis of samples and data from a CSP vaccine clinical trial. Therefore, we have now applied our novel assays and approaches to analyse responses induced by the RTS,S malaria vaccine (which is based on CSP) and their associations with protection from malaria in a phase I/IIa controlled human malaria infection challenge (infection via mosquito bite). These analyses demonstrate that i) neutrophil opsonic phagocytosis was higher in protected than non-protected subjects, and ii) the ability of antibodies to promote binding of FcγRIII and FcγRIIa was also higher in protected than non-protected subjects. Therefore, these data support the potential role for FcγR-mediated mechanisms in vaccine-induced immunity. A manuscript reporting these data has now been prepared and posted online with the pre-print server BioRxiv (reference included in the manuscript). We have discussed these results in the discussion.

It would be interesting to more fully evaluate the strength of association between functional antibodies and naturally-acquired immunity against malaria, and further investigate the conditions required to generate protective responses. To address this question, we require a longitudinal cohort of adults with repeated screening over time, since functional antibodies appear to be too low in children to be contributing to immunity, and the study would need sufficiently frequent sampling and overall sample size to detect a potential protective effect. Unfortunately, we do not have access to such a cohort, and cohorts of this type in adults are extremely rare in malaria research. Most cohort studies have focussed on children and few cohorts have sufficient sampling frequency or sample size of adults to address this question. This should be a focus for future research. We are in discussions with colleagues about the possibility of conducting such a cohort as a future goal.

5. Finally, the relevance of CSP as an antigen is a little confusing. While CSP is an important vaccine antigen, its role in natural immunity is quite unclear. Natural immunity is generally associated with the evolution of broad humoral immune responses, but have been classically associated with blood-stage immunity. Thus, it would be quite interesting to determine whether similar responses observed to the blood stage antigens?

Response: CSP is the major, most abundant, antigen on the surface of the sporozoite, and is the leading vaccine antigen. Naturally acquired immunity to CSP is known to occur, which we find in our study. The reviewer raises a good point that naturally-acquired immunity is thought to largely target blood-stage parasites. Our findings are interesting, and expand our current understanding of acquired immunity, by showing that adults in high transmission areas can acquire substantial functional immunity to sporozoites (Figure 5A-D). A prior study in our Kenyan study population found that high levels of antibodies among adults to sporozoite antigens were associated with reduced risk of infection over time (John CC et al, AJTMH 2005). However, functional mechanisms of these antibodies have not been

previously defined. As suggested by the reviewer, we have now expanded our work by including new data on quantifying antibodies that promote neutrophil phagocytosis of blood-stage merozoites for comparison. This shows for the first time that acquisition of these functional antibodies are acquired earlier against merozoites than against sporozoites (Figure 5).

Minor concerns:

1. It is unclear why a number of experiments to full length CSP were carried out using polyclonal rabbit serum rather than readily available human monoclonal or polyclonal serum (e.g., Figure 1 D, E).

Response: We have used different assays and approaches to demonstrate the phagocytosis effect using different antibody types to help establish the robustness of our assays and finding. We have shown that human antibodies, CSP-specific rabbit polyclonal antibodies (including different regions), and epitope-specific monoclonal antibodies can all promote phagocytosis and FcR interactions. This includes demonstration of activity by antibodies to the NANP central repeat region, which have been shown to be protective in animal models. We have edited our wording to make this clearer in the manuscript.

2. There is a general lack of consistency in the methodology used in Figure 1 that makes comparisons between the panels impossible. Both the target antigen as well as the assay conditions (e.g., serum or no serum) vary from panel to panel.

Response: As noted above, we used different cell types and antibodies and conducted different analyses to answer different questions, as well as to help establish the robustness of our assays and conclusions. We have removed some figures from Figure 1 and edited some of our text to make this clearer and revised the wording of our figure legends. We hope the revised version is better.

3. Given the binding of antibodies to Fcgr3, the inclusion of ADCC activity seems like a natural step. However, the selected method is (using a Fcgr3a-expressing reporter cell line) not a direct ADCC assay and just reports Fcgr3 signaling. This should be clearly specified and not referred to as proof of cytotoxicity.

Response: That is a fair point, and we have addressed this by now including new data from assays using Kenyan adult antibody samples with primary NK cells (which express FcR11a) in a flow-based assay, which quantifies NK activation. Data generated using these new assays suggests a similar finding, that antibodies to CSP can promote NK activity and this correlates with FcR11 binding to antibody-antigen complexes (See new Figures, 2G and 2H). There are currently no published data demonstrating ADCC activity of NK cells against sporozoites; therefore, we believe this is a significant new finding.

4. Neutrophils mediate an array of functions. Beyond phagocytosis, do the functional antibodies trigger additional neutrophil functions (cytokines?)? Do these change with antibody function, FcR usage, IgG titer dependence? Do these downstream inflammatory responses track with protection or inflammation/fever? Enhanced resolution with cohort characteristics may provide enhanced resolution on the immune protective functions of these functional antibodies.

Response: The reviewer rightly notes that neutrophils (and monocytes) can have a range of downstream impacts. Our study aims to define mechanisms and targets of immunity that may clear sporozoites from blood and prevent hepatic infection. To achieve this, mechanisms would need to be functioning within 30-60 mins to be effective. Neutrophil cytokine production, changes in surface molecules, and NET formation all take several hours to occur and would therefore be too late to have a major effect on clearing sporozoites which will typically reach the liver within an hour or so. These extended functions of neutrophils will be much more relevant during blood stage infection that occurs over days or weeks. Therefore, future studies that investigate these neutrophil functions in blood-stage infection would be warranted.

Reviewer #2 (Remarks to the Author):

- Key results: Please summarise what you consider to be the outstanding features of the work.

This work by Feng et al aims to characterise the role of Fc γ R and opsonic phagocytosis in anti-sporozoite-mediated immunity to malaria. Using a range of novel assays to probe samples from a region of medium to high transmission, the role of and interactions between different phagocyte populations and Fc γ Rs is explored. The observation that most opsonisation of sporozoites is neutrophil mediated is novel as is the definition of which Fc γ Rs are utilised. These data are a welcome addition to the field as mechanisms of sporozoite-mediated immunity are poorly understood compared with immunity to blood-stage for example. Although the RTS,S vaccine is the first vaccine to show significant efficacy in the field and targets the sporozoite stage, until recently little immunology has been undertaken to understand the mechanisms of immunity associated with vaccine efficacy.

- Validity: Does the manuscript have flaws which should prohibit its publication? If so, please provide details.

No, overall, the approach is logical, and the experimental methodology is sound.

- Originality and significance: If the conclusions are not original, please provide relevant references. On a more subjective note, do you feel that the results presented are of immediate interest to many people in your own discipline, and/or to people from several disciplines?

The conclusions are original to the best of my knowledge and build on a growing body of work from this and other groups aiming to define protective mechanisms of anti-CSP antibodies. This is a very important area for development of malaria vaccines and is really gathering pace. This study could potentially make an important contribution to that.

- Data & methodology: Please comment on the validity of the approach, quality of the data and quality of presentation. Please note that we expect our reviewers to review all data, including any extended data and supplementary information. Is the reporting of data and

methodology sufficiently detailed and transparent to enable reproducing the results?

This section is where the major weakness of this work lies. Although the methods used are mostly appropriate, the presentation of the data is not and could be improved substantially. I found many of the figures quite difficult to follow and it wasn't always clear what was being shown. In particular, much of data is presented as bar charts, without the number of replicates indicated and so it is impossible to determine the precision of any of the data or how variable the results between replicates were. For the flow-based assay, no information is provided about the total number of events collected. For the OPA, it would be useful to know how many replicate wells were tested or whether only a single well was used. In the legends of the figures 2E and 3A, the authors state "selected individuals are shown as examples" without explaining how the examples were chosen, which could be interpreted as showing only the most favourable data points. In Figure 5, no information is given as to how many participants are included in each age-group. Throughout the paper, p values are presented without defining the test used (although this is stated in the methods section). There was too little information given both on the figures and in the legends to interpret the data without referring back to the text. Some of the figures could be combined together e.g. 4A and B.

Response: All of the points raised by the reviewer have been addressed in the revised manuscript. We respond to specific points as follows:

We apologise for the lack of clarity and missing p values in the previous manuscript. We have now included p values for all figures and tables throughout the paper. In some cases, we have performed new additional experiments to confirm reproducibility and to confirm that differences observed between groups were statistically significant. This is now indicated more clearly in the revised manuscript. For every figure, we have now included p values and indicated the statistical test that was used in the figure legend.

We have reviewed the flow of text throughout the results sections and made revisions and clarifications, and made revisions to some of the figure legends to improve clarity. Some figures were moved to the supplementary data file, and additional new data figures have been included, as described above in the general comments. We have added other specific details requested by all reviewers or that we felt was needed.

Regarding the inclusion of selected individuals shown in the figures, we have tried to clarify this further. We made a selection of individuals that represented the range of responses we observed; these were not selected on the basis of being favourable for our conclusions. For Figure 2E, all data points are shown in Figure 2F. For Fig 2G, all data points are shown in Fig 2H. In Fig 3A, all data points are shown in Fig 3B. We are happy to provide figures in the supplementary file showing all individuals as bar graphs if the reviewer feels that it is important, or we can upload an excel file that includes all data for all individuals.

• **Appropriate use of statistics and treatment of uncertainties:** All error bars should be defined in the corresponding figure legends; please comment if that's not the case. Please include in your report a specific comment on the appropriateness of any statistical tests, and the accuracy of the description of any error bars and probability values.

The statistical tests described are appropriate, however there are numerous references to data in the text that are not supported by a formal analysis e.g. line 378- "Neutrophils had substantially lower opsonic phagocytosis of both CSP-beads and *P. falciparum* sporozoites

in the presence of either FcγRIIIa or FcγRIII blockers (Figure 2A and B)". No indication of whether a statistical test was applied or not.

Response: As noted above, we have now improved our reporting of analysis and differences between groups. The general methods section outlines the statistical tests used. In the figure legends, we have now included p values for all comparisons in all figures, and we have stated the specific statistical test that was used.

• **Conclusions:** Do you find that the conclusions and data interpretation are robust, valid and reliable?

This is very hard to assess because the presentation of the data is not clear enough.

Response: As noted above, we have now improved the presentation of our data, provided p values for all figures and tables, and provided more information about statistical testing and some specific additional detail about methodologies, reagents, and cell types used. Additionally, we have also added new experimental data that support and extend our findings.

• **Suggested improvements:** Please list additional experiments or data that could help strengthening the work in a revision.

• **References:** Does this manuscript reference previous literature appropriately? If not, what references should be included or excluded?

The manuscript is well referenced.

• **Clarity and context:** Is the abstract clear, accessible? Are abstract, introduction and conclusions appropriate?

The abstract and introduction are clear and well-written.

• Please indicate any particular part of the manuscript, data, or analyses that you feel is outside the scope of your expertise, or that you were unable to assess fully.

Specific comments to authors:

1. What was the viability of the cryopreserved *P. falciparum* sporozoites? My understanding is that most are dead, so 50,000 *Pf* sporozoites would be very different to 50,000 freshly dissected *P. berghei* sporozoites. Why were different dyes used to label the two species of sporozoites? Does the cytometer have the same sensitivity for the two dyes, such that the signal is comparable?

Response: We have provided some information in the manuscript about the viability of cryopreserved *Pf* sporozoites (this is established as >85%, and has been published). We note that we have used both cryopreserved *Pf* sporozoites and freshly isolated live Pb-*PfCSP* sporozoites, which have a high viability (Fig 1C,E; 2A; 4A, B, D; Supplementary Figs S4A, B, C)). Therefore, we believe we have addressed any concerns regarding viability of cryopreserved sporozoites. Findings were similar when using the two types of sporozoites.

We have reviewed the text and figure legends to ensure it is clear what sporozoite type we have used in different experiments. We used different dyes across various experiments because of the need to accommodate the use of different cell labelling and antibody dyes. The two dyes used for sporozoites have the same sensitivity.

2. In Figure 1A, the number of replicate wells tested should be stated and the statistical test used stated. In B and C the dotted lines should be described in the legend. In C, relatively few data points are shown to conclude that the correlation between beads and Pf sporozoites was acceptable. The legend mentions high, intermediate and low responses, but only two populations are discernible.

Response: Figures 1A, B, and C in the previous manuscript have now been moved to the supplementary file (Fig S4). P values and statistical tests are now stated in the figure legends, and the wording of the figure legends has been improved. The results section has been revised to address these changes and the reviewer's comments.

3. In Figure 1 D and E, these data would be better represented with line graphs and suitable error bars, I would suggest that for only 6 replicates, a non-parametric average and error would be more suitable.

Response: We appreciate the comments, but we have opted to stay with the bar graph presentation which we find visually effective. We find the data have a near normal distribution and therefore we were comfortable using an ANOVA test. If we use other non-parametric tests, we still see highly significant differences ($p < 0.01$)

4. SF2- the lower panels need to be more clearly labelled as it was hard to determine what they related to (neutrophils on the left and monocytes on the right I think).

Response: As suggested, we have edited these.

5. Fig2A and B, number of replicates is not given and spread of data cannot be assessed, so a different type of graph would be more appropriate. These two panels could be combined. 2E How were the individuals shown selected? 2F could the authors comment on the strength of the correlation with a r of 0.4. It looks like a relatively weak association.

Response: We have added information on replicates and included p values for differences between groups and stated the statistical test used. The figure has been re-organised and edits have been made to the figure legend. We cannot combine Figs 2A and 2B because one is performed with CSP-beads and the other with Pf sporozoites. We have now more clearly stated how individuals were selected to be representative of the range of activities seen. An r value of 0.4 would be considered a moderate correlation in a clinical study where substantial population variance is typically observed. We have now added data on using fresh NK cells in an activation assay (Fig 2G, H), and activity in this assay correlated more highly with FcRIII binding.

6. Figure 3. Again, how were examples selected for 3A? The authors state "Overall 67.3% and 61.5% of samples (n=104) were considered positive for Fc ϵ RIIa and Fc ϵ RIII binding". How was positivity determined?

Response: This is now clarified in the figure legend, and explained in the response above. individuals were selected to be representative of the range of activities seen. Positivity was

classified as response that were greater than the mean+3SD of non-exposed controls, now noted in the methods (Lines 335-7).

7. I have no sense of how reproducible the assay is and how many cells were acquired in order to determine the phagocytosis index? If each data point is just a single well of 5000 cells, how many of those were actually run on the cytometer? Was extracellular fluorescence from sporozoites that were not phagocytosed, but stuck to the outside of cells quenched?

Response: For each sample, we analysed 2500-4000 cells on the flow cytometer, which yields a consistent result (i.e. if the same sample is run on a cytometer repeated times you get the same result). We have previously published methods for opsonic phagocytosis, and we demonstrated that cells are phagocytosed by using a combination of methods including cell lysis approaches, different phagocytosis times and cell surface dyes that only become active in the phagolysosome (Osier, Feng et al, BMC Medicine 2014; Ataide R, et al, PLOS ONE 2010).

8. Figure 5, number of individuals per age-group is not indicated. Statistical tests used are not described.

Response: Further details on the study cohorts used have now been included in the manuscript (page 6). Table S1 in the Supplementary File provides a summary of the clinical features of the study group and the number of individuals per age-group is now indicated in the figure legend (Figure 5). Kruskal-Wallis tests were used and now noted in the figure 5 legend.

9. Figure 6. It is unclear how corrections for multiple analyses was performed, although the authors allude to a correction in the methods section. Given that the multivariate analysis is a much more suitable way to interpret all of these associations, could this figure be moved to the supplementary and Table S4 presented in the main text.

Response: We acknowledge the reviewer's point that the multivariate analysis is an important piece of analysis. We have now moved the previous Table S2 to the main manuscript as Table 1. We believe this is the most important analysis as it shows associations between opsonic phagocytosis with IgG subclasses and FcR binding activity. The other tables summarising multivariate analysis are still included in the Supplementary file (Tables S4 and S6)

Reviewer #3 (Remarks to the Author):

While we have known for decades that CSP-specific antibodies can protect people and mice, at least in part through their impact on sporozoite motility, it remains an open question whether these antibodies also have indirect sporozoite neutralization activity via the Fc portion of antibody. In this study the authors use in vitro phagocytosis assays (with neutrophils or the monocyte cell line THP1), combined with different sources of antisera (rabbit, mouse, or Kenyan adults), and different sources of the target antigen CSP (sporozoites, peptides and recombinant protein on beads) to determine that antisera specific for CSP can enhance the phagocytic uptake of CSP coated beads and antibody-opsonized sporozoites by neutrophils. This is not surprising as most targets in the presence of

antibodies would be taken up in greater amounts by neutrophils in a small well where the optimal pathogen:neutrophil ratio is used and the neutrophils are given sufficient time to recognize and take up the target. What is never addressed is whether this has any relevance to malaria infection.

Response: Our response can be divided into two main points as follows:

i) The significance of higher phagocytosis by neutrophils: We would like to start by clarifying that our findings show that the phagocytosis rate is substantially higher for neutrophils than it is for monocytes. The higher activity of neutrophils is not simply explained by their higher abundance in blood compared to monocytes. This was evaluated across a range of different antibody concentrations and using different bead:phagocyte ratios and demonstrated using freshly isolated sporozoites. The text in the results and discussion has been clarified to help ensure this point is clearer. The assay was not optimised for neutrophil phagocytosis; instead we used assays that we believed would be relevant in vivo based on the following considerations: i) We used a modified whole blood assay that included all leukocytes to better represent in vivo conditions rather than just using isolated cell types or cultured cell lines for assays, which can give misleading results. ii) We used relatively short incubation times (20 mins) to help ensure that the activity we observed occurred in a time period that was relevant to the biological timing of sporozoites entering the blood, and circulating until reaching the liver. Substantial phagocytosis occurs under 20 mins.

An unexpected result was that the phagocytosis rate or activity for neutrophils was much higher than for monocytes. A role for neutrophils in immunity to sporozoites in the blood has not been previously reported or established. We do also demonstrate that monocytes can phagocytose opsonised sporozoites and CSP-beads, and we now provide new data identifying the major monocyte sub-set for this activity (Figure 1D). We extended our investigations to subsequently define the specific FcγRs involved in phagocytosis, which revealed additional unexpected results in demonstrating the major role of FcγRIII. We also show that antibodies can trigger ADCC activity (and we provide new data using primary NK cells; See Figure 2G and 2H). Furthermore, we established that CSP is a major target of these antibodies, including demonstrating that functional antibodies can target all three regions of CSP (N-terminal, central repeat, and C-terminal regions).

ii) The relevance of these findings to malaria infection:

As noted by the reviewer, the importance of antibodies in mediating immunity against sporozoites has been established for several years through clinical association studies and experimental animal models. However, the mechanisms and specific targets of antibody-mediated immunity are not well understood. Few functional antibody mechanisms have been defined and there is no established correlate of protection. As noted by all reviewers, the role of interactions of antibodies with FcRs for immunity to sporozoites has not been previously established, including the role of neutrophils, NK cells, monocyte subsets, specific FcRs involved, antibody properties, or specific epitopes of these functional antibodies. Therefore, these questions form the main focus of our manuscript and represent novel findings.

We believe our findings have clinical relevance for several reasons.

a) We conducted phagocytosis assays using fresh whole blood and freshly collected primary cells rather than cell lines, and we demonstrated effective phagocytosis using different antibody types, different sporozoite sources, and antigen-coated beads to provide different lines of evidence to support our conclusions and demonstrate the robustness of our assays. We conducted assays in a time period (15-20 mins) that is biologically relevant and could

feasibly function in vivo. We observed substantial phagocytosis functional activity when conducting assays in static conditions or using agitation to maintain cells in suspension.

b) Studies in different fields of research (e.g. bacteriology) have generally established that phagocytosis quantified using primary cells in vitro under appropriate conditions is usually indicative that phagocytosis occurs in vivo. Since we demonstrated that opsonic phagocytosis occurred under different conditions and with different reagents in vitro, and we have also established that high levels of phagocytosis can occur in static assays and in assays using agitation, we believe we can be confident that this is a mechanism that occurs in vivo in blood.

c) Importantly, we have now applied our assays to quantify responses induced by the RTS,S malaria vaccine (which is based on the CSP antigen) in a phase I/IIa controlled human malaria infection challenge (via infected mosquito bite). Our analysis found that i) neutrophil opsonic phagocytosis was higher in protected than non-protected subjects, and ii) the ability of antibodies to promote binding of FcγRIII and FcγRIIa was also higher in protected than non-protected subjects. Therefore, these data provide further support for a potential role of FcγR-mediated mechanisms in immunity in vivo. A manuscript reporting these data has been prepared and posted online with the pre-print server BioRxiv. We hope that making these data publically available will provide further confidence in the value of our findings and encourage others to apply these approaches to studies of immunity. We have discussed these considerations and results in the Discussion section.

Additional comments: An exciting and unexpected finding was that antibodies to the N-terminal region have prominent functional activity, which is significant because the N-terminal region is not included in the RTS,S or R21 vaccines that are in clinical trials. To further advance knowledge and inform vaccine development, we have now included new data on the mapping of epitopes in the N-terminal region that can mediate FcγR functional activity (See Figure 4F).

Figures 3, 5, and 6 show that sera from people in a malaria-endemic area are capable of binding to CSP-coated beads and Fc receptors, particularly FcRIII, and be taken up by neutrophils in vitro. The relevance of CSP-coated beads to sporozoites is weak. Sporozoites are one of the fastest eukaryotic cells, capable of easily traversing through phagocytic cells, they are not an inert bead. There are, however, a few experiments with sporozoites, both cryopreserved *P. falciparum* and *P. berghei* expressing PfCSP in Figures 1F&G, 2B, 4A,B&D. None of the experiments with sporozoites have statistics and it looks like the background is between 5-7% phagocytosis (i.e. in the presence of an irrelevant antibody) while the signal is 10-15%.

Response:

i) *Relevance of CSP-coated beads in assays:* The relevance of CSP-beads was established in several different ways. CSP is the major and most abundant antigen on the target of sporozoites and a known target of protective antibodies and is an important target to study. Initially, we demonstrated with a selection of malaria-exposed subjects that phagocytosis of CSP-beads has a high correlation with phagocytosis of sporozoites (now Supplementaryfile, Figure S4). For key findings on opsonic phagocytosis and roles of FcRs, we conducted assays with live intact sporozoites as well as CSP-beads to extend our findings. We confirm that we have demonstrated that neutrophils do demonstrate significant phagocytosis of live intact sporozoites, and that neutrophils have a greater phagocytosis rate than monocytes

when using live sporozoites or CSP-beads (Figure 1). We also confirm that FcR111 and FcR11a mediate phagocytosis when using live sporozoites or CSP-beads (Figure 2), and that antibodies to CSP and specific regions of CSP promote phagocytosis using CSP-beads and sporozoites (Figure 4). All of these results are now supported by p values and the statistical test used is stated. Therefore, we have shown in several different assays and using different antibodies that phagocytosis of live sporozoites, predominantly by neutrophils, does occur. As considered in the Discussion, generating large numbers of sporozoites for extensive assays is challenging because of substantial technical constraints. The use of CSP-beads and FcR-binding assays, which we have established and validated experimentally, provide valuable tools for probing specific functions and evaluating functional antibodies in cohort studies and clinical trials.

ii) Motility of sporozoites: We agree that sporozoites are not an inert bead, and as such we took care to conduct key assays with sporozoites and not just CSP-beads. As noted by the reviewer, sporozoites are motile in the skin, and can also traverse hepatocytes and Kupffer cells in the liver. However, they are not motile in blood; they need a surface or matrix for gliding motility. While there are some limited data on traversal through tissue-resident macrophages, these published data only show traversal of non-opsonised sporozoites. Once sporozoites are opsonised by the right antibodies, any contact with neutrophils, monocytes, and NK cells in blood will lead to engagement of FcRs and activate phagocytosis and ADCC. In our manuscript, we have established that substantial phagocytosis does occur with opsonised live sporozoites, especially for neutrophils which have the greatest activity.

iii) Statistical significance of assays using intact sporozoites: We apologize for the lack of clear indication of the statistical significance for all data in the prior version of the manuscript. This has been corrected in the revised manuscript. All key differences that are described in the manuscript and presented in the figures are statistically significant. P values and the statistical test used are now clearly indicated in the figure legends.

We also note that we can achieve high levels of phagocytosis of live Pb Sporozoites and live Pf sporozoites of up to 50% (meaning that 50% of neutrophils phagocytosed sporozoites within 20 mins) using rabbit antibodies raised against CSP and antibodies from Kenyan adults.

Since PbPfCSP are rodent malaria parasites, the tools exist to test whether any of their findings have any relevance in vivo. Without in vivo experiments, the relevance of the in vitro activity of antibody, particularly when using CSP-coated beads, is not clear. Indeed, the significance of these in vitro findings is questionable given the previous demonstration that adults in malaria endemic areas have no functional pre-erythrocytic stage immunity (TM Tran et al Clin Infect Dis 2013).

Response: Our response to the reviewer's comments is address through discussion of three main points, as follows:

i) In vivo relevance of our findings: Establishing the relevance of findings in vivo in humans can be achieved through several different approaches and different types of evidence are required to build a case to support a causal relationship (E.g. Bradford Hill criteria for causation). We have described above (response to Reviewer 3, Point 1) how we considered the in vivo relevance of our findings. The role of antibodies in immunity to sporozoites has been established over several years in clinical studies and experimental animal models, and antibodies to CSP broadly correlate with protection against malaria with the RTS,S vaccine

(which is based on CSP). While the role of antibodies has been established, the key mechanisms mediating the protective effects of antibodies are poorly understood. Therefore, the main focus of our study was on the mechanisms of FcγR-mediated immunity against sporozoites, which includes understanding specific cell types, FcγR types, antibody targets, and antibody properties. Our findings provide important new insights into antibody-mediated immunity by defining FcγR-mediated mechanisms and targets.

The reviewer comments that '*Since PbPfCSP are rodent malaria parasites, the tools exist to test whether their findings have relevance in vivo*'. We assume by this statement that the reviewer is referring to in vivo in mouse models? To evaluate these FcγR mechanisms in mouse models is very challenging because of major differences between human and mouse FcγRs and IgG subclass activity. In humans, the main activating FcγRs expressed are FcγRIIIa and FcγRIIIa/b by neutrophils, and FcγRI and FcγRIIa (and some FcγRIIIa) by monocytes. In mice, neutrophils express mFcγRIIb, mFcγRIII, mFcγRIV, and monocytes express mFcγRIIb, mFcγRIII, and mFcγRIV. Not only do the FcγR types expressed differ, but their functions and affinity for IgGs also differ between mouse and humans. Further complicating approaches using mouse models is that IgG subclasses and their interactions with FcγRs also differ between humans and mice. In humans, IgG1 is the most prominent response in malaria and vaccination, with IgG3 usually the next most prominent. Both of these subclasses very effectively interact with FcγRI, FcγRIIa and FcγRIIIa/b on monocytes and/or neutrophils. In mice, mIgG1 is the most prominent response, but this does not effectively interact with mouse FcγRs or human FcγRs. Therefore, these differences preclude evaluating the human equivalent of these functional mechanisms in current mouse models of malaria and sporozoite immunity. However, based on findings we present in this manuscript, and our data from analysis of an RTS,S trial, we believe that an investment in developing new rodent models that can represent human IgG interactions with human FcγRs may be warranted as an additional model for vaccine evaluation for the future. This would probably require the use of transgenic mice expressing human FcγRs and the use of human IgG to different targets.

ii) Responses and protection in an RTS,S vaccine clinical trial: To extend our findings and obtain further evidence supporting a role for FcγR-mediated functional antibodies in vaccine-induced protective immunity, we applied our assays to a RTS,S vaccine trial. RTS,S generates antibodies to CSP, which we identified as a major target of FcγR-mediated functional antibodies. Antibodies are the main mediator of protection with RTS,S; however, the functional activity that mediates protection is not well understood. Therefore, there was a good rationale to investigate FcγR-mediated mechanisms with RTS,S. Our exciting findings revealed that opsonic phagocytosis and FcγRIIa and FcγRIIIa/b binding by antibodies was higher in protected than non-protected subjects in a phase I/IIa clinical trial, suggesting these mechanisms are relevant to vaccine efficacy. We have now completed a manuscript describing the results from this study and we have posted the manuscript on the open-access pre-print server BioRxiv. The manuscript (with 6 figures and Supplementary material) is available here: <https://www.biorxiv.org/content/10.1101/851725v1> - Please see Figures 5 and 6 of the BioRxiv manuscript for associations between antibodies and protection.

iii) Findings of the paper by Tran et al Clin Infect Dis 2013: The reviewer suggests that this paper provides evidence that there is no pre-erythrocytic immunity in adults. First, this study did not evaluate any functional antibody activities, therefore it is difficult to compare with our findings. The study included mostly children, with only a small number of adults (stated in the manuscript as n=47 15-25 year olds). There is insufficient power to assess whether adults

were protected against malaria infection, and no analysis of antibody responses. We have shown in our manuscript that substantial functional activity is only observed in adults exposed to high malaria transmission, and is very low in children. A larger study of adults that is suitably powered would be needed to address this question on the role of functional antibodies to CSP or sporozoites in providing protection. We acknowledge that the general consensus in the field is that naturally acquired pre-erythrocytic immunity is generally slow to develop. For this reason we studied adults resident in a high-transmission area of Kenya who were more likely to have substantial antibodies to sporozoites compared to other populations. We have clarified this in the revised manuscript. We would also like to note that other studies have found evidence of pre-erythrocytic immunity in naturally-exposed populations, but none have evaluated FcγR-mediated immunity. For example, John CC et al AJTMH 2005 found in Kenya that high levels of antibodies to sporozoite antigens did correlate with protection from infection in a cohort of adults, but functional activities were not examined. Therefore, a reasonable conclusion might be that in areas of high transmission, some level of protective sporozoite immunity can be achieved by adulthood. We have added text in the revised manuscript to better address this important point (page 27, starting line 614). Studying these responses in detail, as we have done here, may provide further insights to inform vaccine development. Of further interest, Faith Osier and colleagues have presented data at several international meetings on a clinical trial in adult Kenyans that involved experimental infection challenge with sporozoites. These studies showed that significant numbers of adults did not develop blood-stage infection following inoculation of sporozoites, suggesting pre-erythrocytic immunity. These and other data are prompting a re-evaluation of our understanding of naturally-acquired pre-erythrocytic immunity.

Specific Comments:

The most interesting and relevant experiments in this paper have no statistics and the data, with the exception of Fig 4A, do not look significant. Statistics would be needed for Figures 1F, 1G, 2B, 4A, 4B, 4D.

Response: We apologise for the lack of statistics in the previous manuscript. In the current manuscript, p values and statistical testing is provided for all figures and tables.

Controls vary from one experiment to another. Most frequently its unopsonized beads or parasites. This is an important control but more relevant would be adding non-specific sera or antibody. When this was done in Figure 4B (MAB 3D11), the true background looks like its 5 to 7%.

Response: We have included malaria non-exposed control samples, and irrelevant mAbs in some assays to demonstrate the specificity of the assay findings. We hope this is clearer in the revised manuscript.

The concentration of mAb 2H8 used for the opsonization experiment in Fig 4B is not stated in the paper. Do the parasites still move? Are the neutrophils merely mopping up dead immobilized parasites?

Response: We have now included additional data from testing the mAb at different concentrations (Figure 4B). We do not believe the parasites are simply mopping up dead

sporozoites since we established high rates (up to 50%) of phagocytosis using freshly isolated sporozoites that have a high viability and motility rate.

Experiments in which different CSP peptides are immobilized onto beads and then used in phagocytosis and FcR binding assays could be skewed by the amount of peptide that binds to the beads. Frequently this can be sequence dependent. Thus, it is essential to measure this and attempt to have equivalent amounts of peptide per bead for each peptide.

Response: Beads were coated with proteins at a concentration that was many fold higher than the saturating concentration. This would ensure that all beads were coated with a maximum amount of antigen for different antigen constructs. Coating at well above the saturating concentration was performed to achieve binding of proteins at one residue only, aiding the presentation of epitopes and reducing the risk of concealing important epitopes.

The authors demonstrate that the monocyte cell line THP1 is not good for these studies since opsinization/ phagocytosis is inhibited by human serum. The experiments with THP1 should therefore all be moved to the supplemental as beginning the paper with a figure where most of the data is with a cell line that is later shown to be problematic starts the paper off on a weak footing.

Response: This is good point, and we have now moved many of the THP-1 figures to supplementary material, and we have reduced the description and discussion of THP-1 data overall.

Unfortunately there are issues with cryopreserved Pf sporozoites. Once sporozoites are purified, frozen and then thawed, the majority are dead. This is made clear in the clinical trials with these sporozoites, which have necessitated much higher doses to induce immunity than what was originally thought based on irradiated sporozoites delivered by mosquito bite. Freshly dissected PbPfCSP sporozoites are more robust if one is interested in looking at phagocytosis of live sporozoites. For this reason, it would be important to perform the FcR blocking experiment (Fig 2B) and the opsinization/phagocytosis experiments with polyclonal and monoclonal antibodies (Fig 4A&B) with PbPfCSP parasites.

Response: This is an important point and we have therefore added further information to the manuscript. The cryopreserved Pf sporozoites have a high viability rate (>85%) and we have provided a reference that concurs with this (Line 641-3). Furthermore, we have conducted assays using freshly isolated PbPfCSP sporozoites and obtained similar results to using cryopreserved Pf sporozoites or CSP-beads. We would like to clarify that we did not use irradiated Pf sporozoites, but isolated Pf sporozoites that were immediately cryopreserved. The major advantage of using cryopreserved Pf sporozoites is that we can conduct all assays using the same batch of sporozoites, which aids reproducibility. Additionally, using cryopreserved sporozoites is feasible option for future assays to evaluate samples from vaccine trials, whereas using freshly isolated sporozoites to evaluate vaccine responses in clinical trials would be immensely challenging.

References:

Ref 5 does not show that sporozoites are inoculated into a pool of blood. There is no reference I am aware of that shows this. However, the following references show that

sporozoites are inoculated into the skin – Matsuoka et al. Parasitol Int 2002; Sidjanski et al 1997 AmJTropMedHyg.

Response: Apologies that our text was not clear, and we have now revised this. Ref 5 reports that mosquitoes have capillary and blood-pool feeding. We have revised this this point in the introduction.

Refs 11-16 which show that antibodies to CSP inhibit motility should include two recent in vivo studies: Flores-Garcia et al mBio 2018; Aliprandini et al Nat Microbiol 2018.

Response: Thank you for the suggestions. We have now included those references.

Its stated that little is known about functional regions of the CSP. Actually a functional region in the N-terminus was shown in Coppi et al JEM 2005 and 2011, and should be mentioned, especially given their data with the N-terminal peptide.

Response: We agree, the Coppi et al papers were important and relevant publications and we have now referenced these findings in the Discussion.

REVIEWER COMMENTS

Reviewer #1 (Remarks to the Author):

In the paper entitled "Mechanisms of Fc γ -receptor mediated immunity to malaria sporozoites," Feng et al. investigate the Fc region of anti-CSP antibodies may play a role in malaria immunity. The authors start by investigating phagocytosis of CSP-coated beads induced by malaria-exposed Kenyan pooled serum in a whole blood assay and see neutrophil-mediated phagocytosis predominating and go on to confirm differential phagocytosis in immune vs. non-immune serum with EtBr-stained sporozoites. The authors go on to investigate the role of Fc γ Rs in malaria-immune serum triggering THP-1 phagocytosis and ADCC (two methodologies) with Fc γ R blocking antibodies with sporozoites and CSP-coated beads. Next, the authors go on show anti-CSP Abs from Kenyan individuals bind to Fc γ RIIa and Fc γ RIII dimers. The authors investigate reactivity to fragments of CSP with polyclonal rabbit-raised Anti-CSP Ab and pooled samples of Kenyan serum and map the reactivity of the region within the N-terminus of CSP that RpAb recognize. The authors further show that total IgG, Fc γ R binding of anti-CSP antibodies increase with age in a Kenyan cohort and investigate antibody correlates for opsonic phagocytosis adults.

From the first round of reviews, the authors have added a second ADCC/NK assay, addressed the viability of PfSPZ, clarified their methodologies, added more statistical measurements to their figures, have supplied many comparisons of bead-based and sporozoite measurements of Ab activities, added anti-merozoite data to their age-based profiling of Kenyan subjects, added multiple concentration of their single human anti-CSP antibody, and have added references.

Despite extensive revisions and improvements to the paper, the authors have not addressed the major concern about their conclusions that Anti-CSP Fc γ R-mediated Neutrophil phagocytosis is an essential/relevant mechanism of naturally acquired human anti-malarial immunity.

Major Comments:

- 1) The role of Anti-CSP Ab in sporozoite phagocytosis – The authors show in Figure 1 that Kenyan serum induced phagocytosis in neutrophils >> monocytes in their blood assay with anti-CSP Abs. They then repeat this assay with whole sporozoite and redemonstrate phagocytosis of neutrophils >> monocytes. However, the necessity / contribution of Anti-CSP Ab to the whole-sporozoite phagocytosis remains unaddressed. Anti-CSP depletions or co-incubation with an Anti-CSP Ab F'ab that will not induce Fc-mediated functions would help address the contribution Anti-CSP antibodies play. This is essential to establish the relevance of the work to naturally acquired malaria immunity.
- 2) The concerns about the time allowed for phagocytosis for PMNs vs. monocytes remains unaddressed. To address this, the authors could perform doing different length of phagocytosis assays which they did not do or use an in vivo model (mouse or NHP) to investigate the in vivo importance of the different phagocytes. It is important to recognize that the composition of phagocytes in skin and liver is likely different than peripheral blood.
- 3) Experimentation with rabbit pAb serum raised to CSP (Figure 4) demonstrate that neutrophil phagocytosis to the N-terminus of CSP is possible biologically, but does not provide evidence that this is a major determinant of human immunity.
- 4) Phagocytosis and CSP Peptides (Figure 4) It is surprising that phagocytosis of the full length folded CSP protein is lower than both sub-domains peptides – N-term and C-term. I wonder whether we are seeing concentration-dependent bead occupancy affects in addition to possible antibody-specific functional differences.
- 5) Investigating anti-CSP Abs in RTS,S Trial – This is in another manuscript posted on a pre-print server. It is very likely that mechanisms of vaccine-mediated immunity will differ from natural acquired malaria immunity from decades of thousands of exposures in a malaria endemic region. This therefore does not bolster the case that Anti-CSP Fc-driven functions are essential for naturally acquired malaria immunity.
- 6) Age dependence of Anti-CSP Abs and Fc γ R mediate Neutrophil phagocytosis (Figure 5, 6) – The Authors demonstrate that Anti-CSP IgG titers are age dependent. Fc γ R binding and cell

phagocytosis are also anti-body titer. With great variation in IgG titer, it is hard to determine whether the changes in FcγR binding and cell phagocytosis are from the titer or a property of the Abs themselves. It is crucial that authors display their data corrected for titer to make the claim an increase in Ab function tracks with age.

Minor Comments:

1) Serum Free Conditions – The authors incorrectly state that bovine IgGs don't interact with human FcRs. This topic is reviewed in Ulfman et al. <https://doi.org/10.3389/fnut.2018.00052>. They mention their additional experiment where the rate of phagocytosis is significantly lower when human IgGs serum is used is an interesting phenomena that itself should be interrogated.

2) Concerning the expression of FcRs on different immune cells. The authors report that this has already been published in the literature. There is much knowledge about genetic polymorphisms in FcRs in Sub-Saharan Africa and it is enriched in the continent. Little is known about expression or surface receptor-level studies. If the authors are trying to make the argument that Ab-FcR interactions in PMNs are important in human immunity to *P. fal*, understanding if FcγR are present at rest on the pBMCs as previously described.

Reviewer #2 (Remarks to the Author):

The authors have undertaken a significant revision of the manuscript and as a result, the clarity and presentation of the data is much improved. In addition, the specific comment that I made have been addressed both in the rebuttal letter and the manuscript. The figures are now labelled with p values and the justification for statistical methods is sound. I also note that the references have been expanded too to include some important relevant papers. Overall it is much clearer, particularly the methods and analysis and I think this has greatly improved its readability.

Reviewer #3 (Remarks to the Author):

The reviewers have answered most of my concerns.

Response to reviewer comments

Reviewer 2 and 3 did not request any further revisions to our manuscript

Response to Reviewer #1

We thank the reviewer for their comments and we have responded to each point below. To address the points raised we have included significant new data, including data on the effect of depleting CSP antibodies on phagocytosis by naturally-acquired antibodies, demonstrating the phagocytosis activity of a human monoclonal antibody to CSP, evaluating phagocytosis by monocytes and neutrophils at extended time-points, analysis of differences in antibody functional activity between children and adults, evaluating the binding of different Fc γ -receptor alleles, and demonstrating the expression of Fc γ -receptors on neutrophils.

Major Comments:

1) The role of Anti-CPS Ab in sporozoite phagocytosis – The authors show in Figure 1 that Kenyan serum induced phagocytosis in neutrophils >> monocytes in their blood assay with anti-CSP Abs. They then repeat this assay with whole sporozoite and redemonstrate phagocytosis of neutrophils >> monocytes. However, the necessity / contribution of Anti-CSP Ab to the whole-sporozoite phagocytosis remains unaddressed. Anti-CSP depletions or co-incubation with an Anti-CSP Ab F'ab that will not induce Fc-mediated functions would help address the contribution Anti-CSP antibodies play. This is essential to establish the relevance of the work to naturally acquired malaria immunity.

Response: The reviewer raises a good point, indicating that it would be valuable to demonstrate the importance of CSP-specific antibodies for sporozoite phagocytosis promoted by acquired human antibodies. As suggested by the reviewer, we have directly addressed this by performing assays with human antibodies with and without depletion of CSP-specific antibodies. Using human serum antibodies from malaria exposed adults, we found that neutrophil phagocytosis of sporozoites was reduced by 70.3% when CSP antibodies were depleted. These data are shown in **Figure 1F** of the revised manuscript and text explaining the results and methods have been included (from line 433, page 18). Previous studies in the literature have suggested that CSP is the most abundant protein on the surface of the sporozoite.

We also used an additional approach to obtain evidence of the importance of CSP as a target of acquired functional antibodies in human immunity. Prior clinical studies have shown that human experimental infection with live sporozoites induces antibodies to CSP, and recently human monoclonal antibodies (MAbs) expressed by memory B cells from sporozoite-infected adults have been isolated and sequenced. We have now expressed and purified one of these MAbs identified from experimentally infected volunteers (MGG4; IgG1 subclass). In the revised manuscript, we have now included data demonstrating that this MAb can very effectively promote phagocytosis of sporozoites by neutrophils (**see Figure 1G**, and text on page 18). This further supports the conclusion that sporozoite infection in humans leads to antibodies targeting CSP that have functional activity in phagocytosis.

2) The concerns about the time allowed for phagocytosis for PMNs vs. monocytes remains unaddressed. To address this, the authors could perform doing different length of phagocytosis assays which they did not do or use an in vivo model (mouse or NHP) to investigate the in vivo importance of the different phagocytes. It is important to recognize that the composition of phagocytes in skin and liver is likely different then peripheral blood.

Response: To address this point we have included some additional data and revised the text in the appropriate section, and we have added a new figure in the Supplementary File. Data presented in our manuscript show that neutrophils are much more effective at phagocytosis than monocytes. In addition to being more active or effective in phagocytosis, they are also much more abundant than monocytes in blood. For all our experiments, equal time was allowed for phagocytosis by neutrophils and monocytes, and for many assays we used whole blood to best represent *in vivo* conditions. To quantify phagocytosis we used incubation times that would be relevant to mechanisms of action in vivo. Our standard assay allowed phagocytosis to occur for 15-20 minutes. To address the reviewer's question, we have now conducted phagocytosis assays over longer time periods to assess whether there are changes in relative phagocytosis of monocytes and neutrophils. Even extending the time for phagocytosis to 60 minutes, we still observed that phagocytosis by neutrophils was much higher than monocytes (see new data, Supplementary material **Figure S5A**).

We have also added a comment in the discussion that the composition of phagocytes, and their expression of FcγRs, in the skin and liver is different to blood (page 29).

3) Experimentation with rabbit pAb serum raised to CSP (Figure 4) demonstrate that neutrophil phagocytosis to the N-terminus of CSP is possible biologically, but does not provide evidence that this is a major determinant of human immunity.

Response: We appreciate the reviewers' point and we have revised some of the wording in the manuscript to ensure our text is clear. We demonstrate through different means that the N-terminal region of CSP is a target of antibodies that interact with Fcγ-receptors to promote phagocytosis. We presented data using rabbit antibodies to the NT region, and data on the ability of acquired human antibodies to the NT region promote FcγR interactions and phagocytosis. We believe the important point is that the NT region is a target of these functional antibodies – this is important because the NT region it is not included in the leading malaria vaccine based on CSP (RTS,S) or another CSP-based vaccine that is currently in phase 2 trials (R21). Inclusion of NT epitopes in CSP based vaccines may help achieve higher protective efficacy. We believe our findings support the future development of vaccine constructs or approaches that generate antibodies to the NT region as well as other epitopes of CSP. We have made some revisions to the text in the discussion to clarify these points and respond to the reviewers comment (E.g. pages 29, 30).

4) Phagocytosis and CSP Peptides (Figure 4): It is surprising that phagocytosis of the full length folded CSP protein is lower than both sub-domains peptides – N-term and C-term. I wonder whether we are seeing concentration-dependent bead occupancy affects in addition to possible antibody-specific functional differences.

Response: To address this point we conducted some additional phagocytosis experiments using beads coated with the different constructs. In Figure 4C, we have now included data obtained from 6 independent experiments performed in duplicate (and with different neutrophil donors) to obtain a more representative picture. The results still suggest that phagocytosis is higher with the NT construct than the other constructs, but the difference between full-length CSP and the NT construct are not as marked. All antigens were coated onto beads well above the saturating concentration; therefore, the different beads should be equally maximally coated with target antigens. The different activity seen with the different antigen constructs may relate to the epitope density and spatial arrangement of epitopes, and the IgG molecules that bind them, and this can influence interactions with FcγRs. This is an interesting question for future research, and structural biology approaches to better understand the exposure and spatial orientation of epitopes may be valuable. We have made some additions to our discussion to take account of the reviewers' comments (page 32 from line 724).

5) Investigating anti-CSP Abs in RTS,S Trial – This is in another manuscript posted on a pre-print server. It is very likely that mechanisms of vaccine-mediated immunity will differ from natural acquired malaria immunity from decades of thousands of exposures in a malaria endemic region. This therefore does not bolster the case that Anti-CSP Fc-driven functions are essential for naturally acquired malaria immunity.

Response: To support the role of antibodies to CSP and Fcγ-receptor mediated immune mechanisms, we have now added additional data to the revised manuscript, as outlined above (point 1). i) We showed that depletion of CSP antibodies from naturally-acquired antibodies of malaria-exposed individuals greatly reduced phagocytosis activity (**Figure 1F**). ii) Monoclonal Abs from B cells isolated from individuals exposed to sporozoite infections have been recently shown to frequently target CSP. We have expressed one of these anti-CSP mAbs and shown that it effectively promotes phagocytosis of sporozoites by neutrophils (**Figure 1G**).

In our manuscript we have established novel concepts in understanding mechanisms and targets of immune responses against sporozoites, and developed new assays and approaches to quantify these responses in clinical studies and vaccine trials. Initial application of these concepts and approaches to a phase 1/2a vaccine trial of RTS,S (based on the CSP antigen) found that FcγR functional antibodies correlated with vaccine efficacy. We believe that these findings broadly support the role of functional mechanisms in immunity to sporozoites and demonstrate the translational relevance of our new findings. However, there is much still to be investigated to understand naturally-acquired and vaccine-induced immunity. We hope that the publication of this paper will provide a catalyst for future research into these mechanisms in immunity. In further acknowledgement of the reviewer's comments, we have also included some additional discussion in the revised manuscript (end of page 32)

6) Age dependence of Anti-CSP Abs and FcγR mediate neutrophil phagocytosis (Figure 5, 6) – The Authors demonstrate that Anti-CSP IgG titers are age dependent. FcγR binding and cell phagocytosis are also anti-body titer. With great variation in IgG titer, it is hard to determine whether the changes in FcγR binding and cell phagocytosis are from the titer or a property of the Abs themselves. It is

crucial that authors display their data corrected for titer to make the claim an increase in Ab function tracks with age.

Response: This is a good point and we have now added new analysis and a figure (**Figure 5G**) that addresses this – we agree that it is valuable to understand whether the higher functional activity among adults results only from the higher concentration of CSP-specific IgG or also reflects a higher functional potential of antibodies among adults. To address this, we analysed the functional activity of antibodies relative to IgG reactivity against CSP as suggested by the reviewer. To do this we excluded individuals with no IgG to CSP, and we calculated phagocytosis functional activity relative to IgG level for each individual. We have then plotted the median (and range) of relative functional activity for children versus adults (we have explained the methodology and analysis in the revised manuscript). Interestingly, this demonstrates that the relative functional activity is significantly higher in adults compared to children (**Figure 5G**). We believe these are the first data to reveal a difference in the functional potential of antibodies to sporozoites between adults and children.

Minor Comments:

1) Serum Free Conditions – The authors incorrectly state that bovine IgGs don't interact with human FcRs. This topic is reviewed in Ulfman et al. <https://doi.org/10.3389/fnut.2018.00052>. They mention their additional experiment where the rate of phagocytosis is significantly lower when human IgGs serum is used is an interesting phenomena that itself should be interrogated.

Response: We acknowledge that our wording on this point needed to be clearer, and we have edited the text to clarify the meaning in our revised manuscript. Phagocytosis by THP-1 cells is mainly mediated by FcγRI (see Figure 2C). It is known that non-specific IgG in human serum will bind FcγRI, which may explain why inclusion of human serum in assays with THP-1 cells results in lower phagocytosis (because it is partly blocking FcRI mediated functions). For IgG to engage FcγRII or FcγRIII, it needs to be in an antigen-antibody complex. Therefore, non-specific IgG in human serum will not inhibit interactions of FcγRII or FcγRIII with specific IgG bound to an antigen.

Bovine IgG can bind human FcγRII, but not FcγRI or FcγRIII. This may explain why the inclusion of FCS or FBS in assays with THP-1 cells does not inhibit human antibody mediated phagocytosis by THP-1 because this is mediated by FcγRI.

Therefore, in conclusion, we believe our hypothesis is a plausible explanation for the difference between THP-1 activity when using human serum or FCS/FBS in the assay media. We have edited the text to make this point clearer.

2) Concerning the expression of FcRs on different immune cells. The authors report that this has already been published in the literature. There is much knowledge about genetic polymorphisms in FcRs in Sub-Saharan Africa and it is enriched in the continent. Little is known about expression or surface receptor-level studies. If the authors are trying to make the argument that Ab-FcR interactions in PMNs are important in human immunity to *P. fal*, understanding if FcγR are present at rest on the PBMCs as previously described.

Response: There are two points here, which we will answer in turn.

i) *Polymorphisms in Fcγ-receptors*: Our data indicate that FcγRIIa and FcγRIII are the most important receptors in phagocytosis (and FcγRIII is important for NK cell ADCC activity). FcγRIIa and FcγRIII occur in populations as different alleles. We are pleased to now include additional new data using different FcγRIIa and FcγRIII alleles (**see Figure 3C and D**). We have established that antibodies to CSP effectively promote binding of both alleles of FcγRIIa (R131, H131) and both alleles of FcγRIII (F158, V158). There was a high correlation of FcγR binding between the different alleles; $\rho=0.8371$ for FcRIIa alleles, and $\rho=0.78$ for FcγRIII. This suggests that although different alleles of FcγRs occur in populations, these polymorphisms do not substantially impact the functional mechanisms we report in this manuscript. We believe these are the first data evaluating the antibody binding activity of different FcγR alleles in human malaria.

ii) *Expression of Fcγ-receptors*: The expression of Fcγ-receptors on human leukocytes is well established. However, we acknowledge that it is difficult to identify papers that clearly report FcγR expression profiles given the very large number of papers published on FcγRs. We refer the reviewer to an excellent paper published this year by Kerntke C et al, *Frontiers in Immunology*, 2020. This paper provides a very clear picture of the FcγRs expressed by different human leukocytes. What is particularly valuable about this paper is that they quantified the copy number of each FcγR on each cell type. Figure 4 in the paper is very clearly presented, showing the FcγR expression profiles. We have pasted a copy of the figure below.

The full paper can be found here: <https://doi.org/10.3389/fimmu.2020.00118>

This paper shows that FcγRIIa and FcγRIII are abundantly expressed on neutrophils, whereas there is little or no expression of FcγRI, or the inhibitory receptor FcγRIIb. Classical monocytes (the most abundant population) express FcγRI and FcγRIIa, and non-classical monocytes express FcγRII and FcγRIII, but also some inhibitory FcγRIIb.

We have confirmed that we observe this in our laboratory. In the revised manuscript we have now included a figure in the supplementary material showing FcγR expression on neutrophils and monocytes (Figure S5C and D). We already have a figure in the supplementary material showing the expression of FcγRs on monocytes and the gating strategy to identify the three main monocyte subsets of classical, non-classical, and intermediate (currently Figure S5B).

Kerntke C et al, 2020: Figure shows the quantification of Fc γ -receptors on human peripheral blood leukocytes. Box plots show copies of (A) Fc γ RI, (B) Fc γ RIIa, (C) Fc γ RIIb, and (D) Fc γ RIIIa/b on indicated leukocyte subsets

REVIEWER COMMENTS

Reviewer #4 (Remarks to the Author):

Several of the concerns raised during initial review were addressed by the authors. However, the following concerns remain:

1) In lines 507-509, the authors state: "This suggested that antibodies to the NT region had a higher potential to engage FcγRs than the central repeat and CT regions." However, no statistical analysis is performed in Fig. 4E. Is this stated difference 1) statistically significant, and 2) biologically meaningful? Indeed, in Fig. 4C where the phagocytosis index is measured presumably from the same antibodies from a pool of Kenyan adult donors, there doesn't appear to be any statistically significant differences for antibodies recognizing the different epitopes. The ELISA titers to different epitopes (NTD, repeat and CTD) should also be measured and reported for antibodies elicited in natural infection.

In fact, that this Results section and Fig. 4 presents data interchangeably from natural infection and rabbit immunization with recombinant PfCSP is very confusing. It is this Reviewer's suggestion that these two datasets be clearly separated, and be presented and discussed separately. This would provide the reader with a clearer view of the strength of the phagocytosis data for antibodies targeting the different domains coming from 1) the biologically-relevant context of human natural infection, and 2) the artificial context of rabbit antibodies elicited by immunization with recombinant PfCSP domains. Overall, it is this Reviewer's opinion that the argument of NTD-directed antibodies being of higher quality than those targeting other PfCSP domains is weak based on the data presented and should be down-played.

2) In the data presented in Figure 4D, it is unclear why rabbit antibodies against the repeat region weren't designed in the experiment to be obtained and their ability to promote phagocytosis of PfCSP-P. berghei sporozoites measured. What is the relative contribution of repeat-specific antibody to phagocytosis, given the importance of the repeat region of PfCSP as an antibody target?

3) In Figure 4F, the amino acid sequences of the reactive peptides (17-20) should be provided.

4) In the Discussion, lines 601-604 and 655-659 are very similar in concept. In fact, the Discussion would benefit from a more focused re-organization and from being more succinct; indeed, it currently runs 7 pages long, and reads more like a commentary.

5) The ability of antibody to trigger the degranulation of NK cells is reported, but the relative contribution of NK cells to ADCC in this context is not discussed. Given the focus on neutrophil-mediated phagocytosis, and the evidence presented for FcγR binding of receptors present on NK cells, it would be valuable to add discussion on the relative role of NK cell ADCC.

6) Binding to different FcγR alleles is investigated in the manuscript, but is missing from the methods. A mention of the prevalence of these different alleles in the human population used in the study would also be beneficial.

7) Are the samples in Fig 1F paired? (ie. depleted and undepleted from the same sera pool completed simultaneously? If so, it would be valuable to show them as such to allow for a direct comparison of the depletion)

8) Figure 2 F - G would benefit from Y axis units. Is the reporter readout RLU?

9) There's a typo in line 376 - "largh" should be "large"

Response to the reviewer's comments

Note: Reviewer's comments are included in blue text. Comments were provided by Reviewer 4 and we have responded below to all points raised by the reviewer.

Reviewer #4 (Remarks to the Author):

Several of the concerns raised during initial review were addressed by the authors. However, the following concerns remain:

1) In lines 507-509, the authors state: "This suggested that antibodies to the NT region had a higher potential to engage FcγRs than the central repeat and CT regions." However, no statistical analysis is performed in Fig. 4E. Is this stated difference 1) statistically significant, and 2) biologically meaningful? Indeed, in Fig. 4C where the phagocytosis index is measured presumably from the same antibodies from a pool of Kenyan adult donors, there doesn't appear to be any statistically significant differences for antibodies recognizing the different epitopes. The ELISA titers to different epitopes (NTD, repeat and CTD) should also be measured and reported for antibodies elicited in natural infection.

RESPONSE: We have now revised the manuscript to address these comments from the reviewer. In the previous version of the manuscript, statistical testing and p values were included in the figure legend for every figure. For Figure 4E in the previous manuscript, it was stated in the figure legend that '...FcγR binding to the NT region was significantly higher compared other constructs ($p < 0.01$)'. For Figure 4C in the previous manuscript, the differences between the different protein regions were not as clear and the p value was indicated for comparisons across the groups. In the revised manuscript, we have now modified the text in the figure legend to make this clearer (now Figure 4A and B; Lines 1018-1030) and we have re-ordered the sequence of figures in Figure 4 to address the reviewer's point below. To help ensure this point is clear, we have made some additional edits in the results section where we describe these findings (Lines 488-496).

Regarding the second point on the ELISA titres to the different regions, all phagocytosis and Fcγ-receptor binding results are expressed relative to ELISA titres, as explained in the methods and noted in the results. We have now added the ELISA titre for the human samples using in Figure 4 as Supplementary Figure S8A. We have revised the text in the figure legends to ensure this is clear.

In fact, that this Results section and Fig. 4 presents data interchangeably from natural infection and rabbit immunization with recombinant PfCSP is very confusing. It is this Reviewer's suggestion that these two datasets be clearly separated, and be presented and discussed separately. This would provide the reader with a clearer view of the strength of the phagocytosis data for antibodies targeting the different domains coming from 1) the biologically-relevant context of human natural infection, and 2) the artificial context of rabbit antibodies elicited by immunization with recombinant PfCSP domains. Overall, it is this Reviewer's opinion that the argument of NTD-directed antibodies being of higher quality than those targeting other PfCSP domains is weak based on the data presented and should be down-played.

RESPONSE: We thank the reviewer for this suggestion and we have subsequently reorganised the figures and the text in the results section of the manuscript. We now first describe our findings on the activity of human antibodies (now Figures 4A and B), and then describe the functional activities of antibodies raised in experimental animals (in Figures 4C, D, E, and F). We have also made some minor edits to the wording in this section to improve the clarity

These changes have been made in the results section titled 'Regions of CSP targeted by functional antibodies' (starting on Line 480).

We acknowledge the reviewer's comments about down-playing or moderating our discussion of the significance of the higher functional activity of antibodies to the NT region. We have revised the wording in the manuscript where appropriate and ensured that our reporting of findings is clear. We found that human antibodies to the NT region had greater potential to engage Fc γ -receptors than antibodies to the other regions of CSP, and antibodies to the NT region were more effective at promoting phagocytosis than antibodies to the CT region. While these results are promising, future studies will be needed to further quantify the protective potential of antibodies to different regions or epitopes of CSP.

2) In the data presented in Figure 4D, it is unclear why rabbit antibodies against the repeat region weren't designed in the experiment to be obtained and their ability to promote phagocytosis of PfCSP-P. berghei sporozoites measured. What is the relative contribution of repeat-specific antibody to phagocytosis, given the importance of the repeat region of PfCSP as an antibody target?

RESPONSE: In response to the reviewer's point here (and their earlier point) we have now clarified this section of the results in our manuscript. In our study we did evaluate functional antibodies that were raised against the three different regions of CSP and also provided data from analysis of human antibodies. We used a monoclonal antibody to the NANP-repeat region, and rabbit antibodies raised against the NT and CT regions. Our data show that antibodies raised against all three regions can promote phagocytosis; our data with human antibodies also demonstrated that all three regions are targets of functional antibodies. The relative role of repeat-specific antibodies in phagocytosis and Fc γ R interactions is shown in Figures 4A, 4B, and 4D. We have now reorganised Figure 4 and the relevant text (Lines 480-523) to make this whole section clearer, by first presenting the data with human antibodies (now Figure 4A and B) and then presenting data using antibodies raised in animals (Figures 4C-F).

3) In Figure 4F, the amino acid sequences of the reactive peptides (17-20) should be provided.

RESPONSE: Thank you for noting this. We have now added a supplementary figure (Figure S8B) which shows the amino acid sequence for all peptides in the array and the relative antibody reactivity to each peptide. In the results section of the manuscript (Line 523), we have also stated the amino acid sequence that was reactive to IgG.

4) In the Discussion, lines 601-604 and 655-659 are very similar in concept. In fact, the Discussion would benefit from a more focused re-organization and from being more succinct; indeed, it currently runs 7 pages long, and reads more like a commentary.

RESPONSE: We appreciate the reviewer's suggestions and have reviewed the discussion to reduce the length where possible. We have removed the repetition of similar points, as suggested. The length of the discussion has partly resulted from the need to address several points raised by various reviewers of the manuscript during the revision process, including requests from different reviewers to include specific points in the discussion. However, we have been able to reduce the length to ~1700 words, which we feel is reasonable relative to other published papers and the amount of data we have included in the manuscript.

5) The ability of antibody to trigger the degranulation of NK cells is reported, but the relative contribution of NK cells to ADCC in this context is not discussed. Given the focus on neutrophil-mediated phagocytosis, and the evidence presented for FcγR binding of receptors present on NK cells, it would be valuable to add discussion on the relative role of NK cell ADCC.

RESPONSE: We have now added some discussion on this point in the manuscript (Lines 633-636). Considering the reviewer's point above about reducing the length of the discussion, we have kept the discussion of NK cells brief. Of note, NK cells comprise only 1-5% of leukocytes in blood suggesting they would play a less prominent role in immunity against sporozoites than neutrophils, which comprise 50-70% of leukocytes.

6) Binding to different FcγR alleles is investigated in the manuscript, but is missing from the methods. A mention of the prevalence of these different alleles in the human population used in the study would also be beneficial.

RESPONSE: We have now mentioned the different FcγR alleles in the methods (line 312). We have now included a reference to a paper that reported the prevalence of the different FcγRIIIa and III alleles in western Kenya, which is where we conducted our studies of adults and children (lines 474-475). We have not included much discussion on this point because we did not observe any substantial difference in activity between the allelic forms (line 671-672), and we are mindful of the reviewer's comments on the length of the discussion.

7) Are the samples in Fig 1F paired? (ie. depleted and undepleted from the same sera pool completed simultaneously? If so, it would be valuable to show them as such to allow for a direct comparison of the depletion)

RESPONSE: The sample used in Figure 1F was a pool of antibodies from malaria-exposed adult donors. We have shown the phagocytosis activity promoted by the antibody pool before and after depletion of CSP-antibodies, including data from repeated assays. We have reviewed this figure and do not feel that plotting the figure differently would make it clearer. The data clearly show that phagocytosis activity was much lower with the CSP-Ab depleted sample. Statistical significance is indicated in the figure and we have stated the statistical test used in the figure legend. All data points are shown in the figure, which is a requirement of Nature Publishing.

8) Figure 2 F - G would benefit from Y axis units. Is the reporter readout RLU?

RESPONSE: Thank you for picking up that omission. We have now included the parameters measured in these assays (shown on the Y-axis) in the figure legend.

9) There's a typo in line 376 – "largh" should be "large"

RESPONSE: This has been corrected.

REVIEWERS' COMMENTS

Reviewer #4 (Remarks to the Author):

The authors have adequately addressed all concerns raised during initial review. Particularly, addition of ELISA titers for the human samples in Supp Fig 8A is insightful. The new separation for the activity of human antibodies (Fig 4A-B) and animal-elicited antibodies (Fig 4C-F) also facilitates reading of this Results section. Finally, the Discussion has been improved for its flow and brevity.

RESPONSE TO REVIEWERS' COMMENTS

Reviewer #4 (Remarks to the Author):

The authors have adequately addressed all concerns raised during initial review. Particularly, addition of ELISA titers for the human samples in Supp Fig 8A is insightful. The new separation for the activity of human antibodies (Fig 4A-B) and animal-elicited antibodies (Fig 4C-F) also facilitates reading of this Results section. Finally, the Discussion has been improved for its flow and brevity.

Response: We thank the reviewer for the comments. The reviewer has no further issues or revisions for us to address.